# Compounding future escalation of emissions- and irrigation-induced increases in humid-heat stress

Yi Yao [1,2] ✉, Yusuke Satoh [3], Nicole van Maanen [4], Sabin Taranu[1], Jessica Keune [5], Steven J. De Hertog[1,6], Seppe Lampe[1], David M. Lawrence [7], William J. Sacks[7], Yoshihide Wada [8], Agnès Ducharne [9], Benjamin I. Cook[10,11], Sonia I. Seneviratne [2], Laibao Liu [12,13], Jonathan R. Buzan[14], Jonas Jägermeyr[10,15,16] & Wim Thiery [1]

Irrigation has been investigated as an important historical climate forcing, but there is no study exploring its future climatic impacts considering possible changes in both extent and efficiency. Here, we address these issues via developing irrigation efficiency scenarios in line with the Shared Socio-economic Pathways (SSPs), implementing these in the Community Earth System Model, and applying them to generate projections over the period 2015–2074. We project that annual irrigation water withdrawal decreases under SSP1-2.6 (from ~2100 to ~1700 km³ yr⁻¹) but increases under SSP3-7.0 (to ~2400 km³ yr⁻¹), with some new irrigation hot spots emerging, especially in Africa. Irrigation is projected to reduce the occurrence of dry-heat stress under both scenarios, but cannot reverse the warming trend due to greenhouse gas emission (e.g., increasing from ~90 to around 600 and 1200 hours yr⁻¹ in intensely irrigated areas, under two scenarios). Moreover, moist-heat extreme event frequency increases more substantially (by ≥1600 hours yr⁻¹ under SSP3-7.0 in tropical regions), and irrigation further amplifies the hours of exposure (for example, by ≥100 hours yr⁻¹ in South Asia), thereby raising the risk of moist-heat-related illnesses and mortality for exposed communities. Our results underscore the importance of reducing greenhouse gas emissions, limiting irrigation expansion and improving irrigation efficiency to preserve water resources and decelerate escalating exposure to dry- and moist-heat stress.

With increasing population size and improving life quality, the area equipped for irrigation (AEI) has expanded substantially in the last century, from less than $1 \times 10^6$ km² at the beginning of the 20th century to ~3 × 10⁶ km² in 2015[1,2]. Expanding irrigation affects local climate conditions by changing the water and energy balance, making irrigation an increasingly important historical climate forcing[3–6]. Owing to the projected growth in food and

fibre demand[7], irrigation will likely continue to expand under most future scenarios. However, the few existing studies exploring the impacts of irrigation under future scenarios[8–10] all assume that there are no changes in AEI or irrigation water flux relative to the present-day.

The water applied for irrigation increases surface and soil moisture in agricultural fields, causing evaporative cooling of the

---

near-surface air[4,11]. This cooling impact is generally more pronounced during heat extremes and, therefore, irrigation can alleviate these extremes[5,11–13]. However, evaporated water also increases near-surface air humidity, which in turn dampens humans' ability to cool down through sweating. As a result, irrigation can actually reduce the thermal comfort of people despite its cooling benefit[6,14,15]. Yet, existing studies exploring irrigation's climatic impacts under future scenarios focused solely on temperature, which may induce an over-optimism of irrigation's cooling benefits for human beings.

With expanding AEI, irrigation water demand is also expected to increase, leading to water scarcity, soil salinisation, and other environmental issues[16–18]. Thus, efficient irrigation techniques, like drip and sprinkler irrigation, are increasingly required to replace traditional flood irrigation systems. These techniques differ in the amount and way of applying water and therefore have varying impacts on the surface water cycle, energy balance, and climate[19,20]. For example, remote sensing data show that water-saving irrigation has attenuated the cooling impacts of irrigation on surface temperature by 0.48 °C dec[−1] in South Xinjiang, China[21], and idealised simulations reveal that switching from traditional flood to drip irrigation could slightly decrease wet-bulb temperature in North India[22]. Considering transition pathways in irrigation techniques is therefore important when projecting irrigation-induced impacts on heat extremes. However, there is currently no global-scale dataset on projected shifts in irrigation methods. Previous efforts therefore, relied on an idealised annual growth in irrigation efficiency[23,24], or remained limited to a conceptual level[25].

Recent efforts have been made to include different irrigation techniques in the land component of Earth system models[19,20,26], and a dataset on spatial information about global irrigation techniques share (ITS) exists[27]. These developments enable us to account for irrigation techniques in Earth system simulations. However, these newly developed irrigation modules have so far only been applied in land-only simulations, thereby ignoring irrigation-induced land-atmosphere feedbacks to the changes in surface water and energy budgets. Moreover, these models so far cannot implement multiple irrigation techniques for one crop type in one grid cell, nor update the fraction of different techniques to account for temporal changes.

To address these issues, we first develop a dataset of transient ITS under a range of Shared Socioeconomic Pathways-Representation Concentration Pathways (SSP-RCPs: see Methods)[28,29]. Two SSP-RCPs, SSP1-2.6 (the green road, with low challenges for both mitigation and adaptation) and SSP3-7.0 (the rocky road, with both high challenges), are selected. Building on a country classification framework[30] and on available socioeconomic and climate projections, we estimate the fraction of irrigation techniques at the grid scale. A basic assumption is that countries with higher socio-economic capacities and drier climate conditions have more motivation to upgrade their irrigation system, which varies among scenarios. We subsequently implement these transient irrigation pathways in the Community Earth System Model version 2 (CESM2) with some developments that enable the model to update ITS information yearly (see Methods).

In this study, three irrigation techniques are considered for most crop types: drip irrigation, sprinkler irrigation, and flood irrigation. For rice, paddy irrigation is applied by default and is accounted for as flood irrigation in the generated ITS dataset. We then perform transient CESM2 simulations for the time period 1985–2074 under SSP1-2.6 and SSP3-7.0, once with and once without transient irrigation (three ensemble members for each experiment). This approach allows us to unravel the interacting impacts of anthropogenic climate forcing and irrigation transition pathways on local heat stress. Here, we focus on changes in irrigation water withdrawal (IWW) and irrigation-induced impacts on dry- and moist-heat stress.

## Results

### Irrigation water withdrawals highly depend on projected irrigation expansion and efficiency

The Land Use Harmonization version 2 (LUH2) dataset[31] shows that under SSP1-2.6, the global AEI remains nearly constant at around 2.7–2.8 × 10[6] km[2], with a rapid increase in the projected use of efficient irrigation techniques according to our dataset (drip and sprinkler accounting for ~ 2/3 in 2100, up from ~1/9 in 2015) (Fig. 1a). Under SSP3-7.0, projected irrigated land[31] increases steadily, reaching more than 4 × 10[6] km[2] by the end of the century, with a smaller fraction of drip and sprinkler techniques (less than 1/2 in 2100) (Fig. 1b).

Based on the spatial distribution of AEI and ITS (Figs. 2, 3, S1, and S2), we select 13 IPCC reference regions[32] with high irrigation activity or with varying changes in AEI under two scenarios (Fig. 4a), and pool them into three groups (Table S1). Group 1 includes regions (solid line in Fig. 2a) with higher socio-economic capacities, like Central North America and East Asia, enabling AEI to remain nearly constant and efficient irrigation to grow rapidly under both scenarios (slightly faster under SSP1-2.6, see Fig. 1c, d and S3). Group 2 consists of the regions (dashed line in Fig. 2a) with large AEI but lower socio-economic capacities in present day, such as South Asia and West Central Asia. In Group 2, AEI remains relatively constant under SSP1-2.6 but expands rapidly under SSP3-7.0 (Fig. 1e, f, and S4). The regions in Group 3 (dotted line in Fig. 2a) have almost no AEI historically and under SSP1-2.6, but their AEI experiences substantial expansion under SSP3-7.0, making them irrigation hot spot regions by the end of the century. These future hotspot regions, for example, include West and South Africa (Fig. 1g, h, and S5).

IWW is highly related to spatial and temporal changes in both AEI and ITS (Fig. 4). From 1985 to 2014, North India was the most intensely irrigated area with simulated IWW surpassing 500 mm yr[−1] in some grid cells (over the entire 0.9° × 1.25° grid cell area). Other heavily irrigated areas included East China and Central USA, where simulated IWW also exceeds 300 mm yr[−1] in several grid cells (Fig. 4a). Globally, IWW ranges between 1700–2000 km[3] yr[−1] during the historical period (in historical simulations, AEI and ITS are static at the level of the year 2010), and IWW projections start from around 2100 km[3] yr[−1] in the year 2015 (Fig. 4d).

IWW is projected to decrease in many grid cells over Central North America and East Asia under SSP1-2.6 (2045–2074) compared to the historical period (1985–2014), where the reduction exceeds 50 or even 100 mm yr[−1] locally (Fig. 4b). Global IWW decreases slightly to ~1700 km[3] yr[−1] by 2074 (Fig. 4d), with the regional values falling from ~400 to ~300 km[3] yr[−1] in Group 1 (Fig. 4e and S6a–c), from ~1300 to ~1100 km[3] yr[−1] in Group 2 (Fig. 4f and S6d–h), and remaining constant at around 50 km[3] yr[−1] in Group 3 (Fig. 4g and S6i–m).

Under SSP3-7.0 (2045-2074), the reduction in IWW persists in Central North America and East Asia, but new irrigation hot spots appear across Africa, and IWW increases in some traditional irrigation hot spots like in South Asia (Fig. 4c). After 2015, IWW increases continuously to ~2400 km[3] yr[−1] under SSP3-7.0 (Fig. 4d). IWW shows different trends across the three region groups due to the varying socioeconomic developments in each group. In Group 1, IWW decreases to less than 300 km[3] yr[−1] (Fig. 4e and S6a−c), owing to a steady AEI and improvements in ITS (Fig. 1c, d and S3). In Group 2, IWW increases to ~1400 km[3] yr[−1] (Fig. 4f and S6d–h), mainly attributed to the expanding AEI (SSP3-7.0) (Fig. 1e, f and S4). In Group 3, a sharp increase from 50 to 400 km[3] yr[−1] in IWW is projected (Fig. 4g and S6i−m) due to a rapid AEI expansion (Fig. 1g, h and S5).

### Divergent impacts of irrigation on dry- and moist-heat extremes

Changes in IWW affect irrigation-induced impacts on surface climate, and here we focus on dry- and moist-heat extremes. We define dry-heat extremes as events when the 3-hourly, 2-metre air temperature ($T_{2m}$) exceeds the 99th percentile (that is, about the 30 warmest 3-hourly

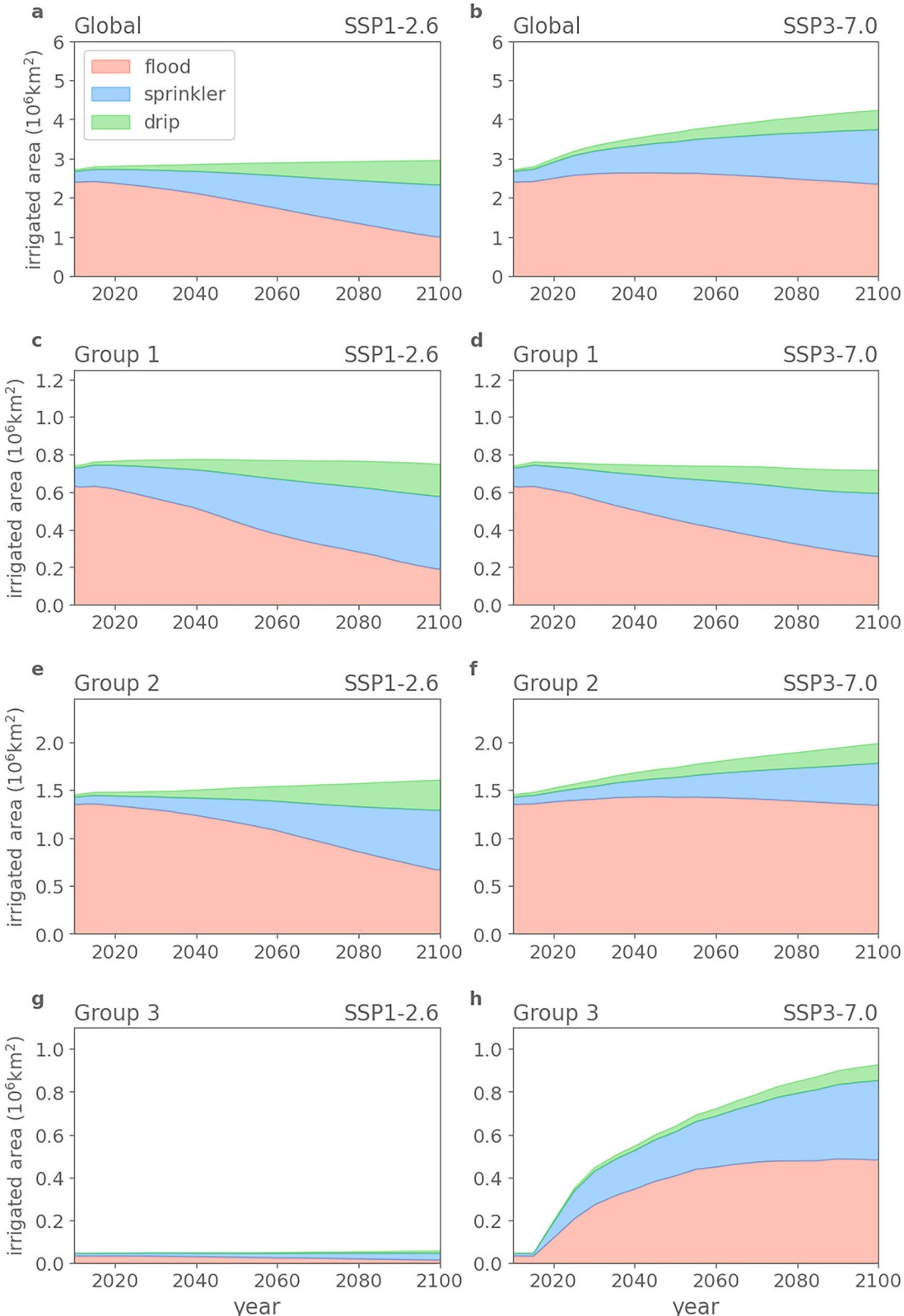

**Fig. 1 | Regionally varying changes in irrigation extent and irrigation techniques.** Global (**a**, **b**) and regional (**c–h**) evolution of areas equipped for irrigation (AEI) with different irrigation techniques under SSP1-2.6 (**left column**) and SSP3-7.0 (**right column**). IPCC reference regions belonging to each group are listed in Table S1, and their locations are visualised in Fig. 4a. AEI and irrigation techniques share (ITS) of individual regions are shown in Fig. S3–5.

blocks, or 90 hours of a year) during the period 1985–2014 in the pooled simulations without irrigation (Fig. 5a). Without irrigation, South Asia, West Central Asia, and the southwestern part of the Sahara exhibit the most extreme temperatures, with the 99th percentile value of $T_{2m}$ reaching 45 °C over large areas. Other areas with high absolute values of dry-heat extremes include Central North America, North Central America, and some regions in South America and Australia. Analogous to dry-heat extremes, we define moist-heat extremes as

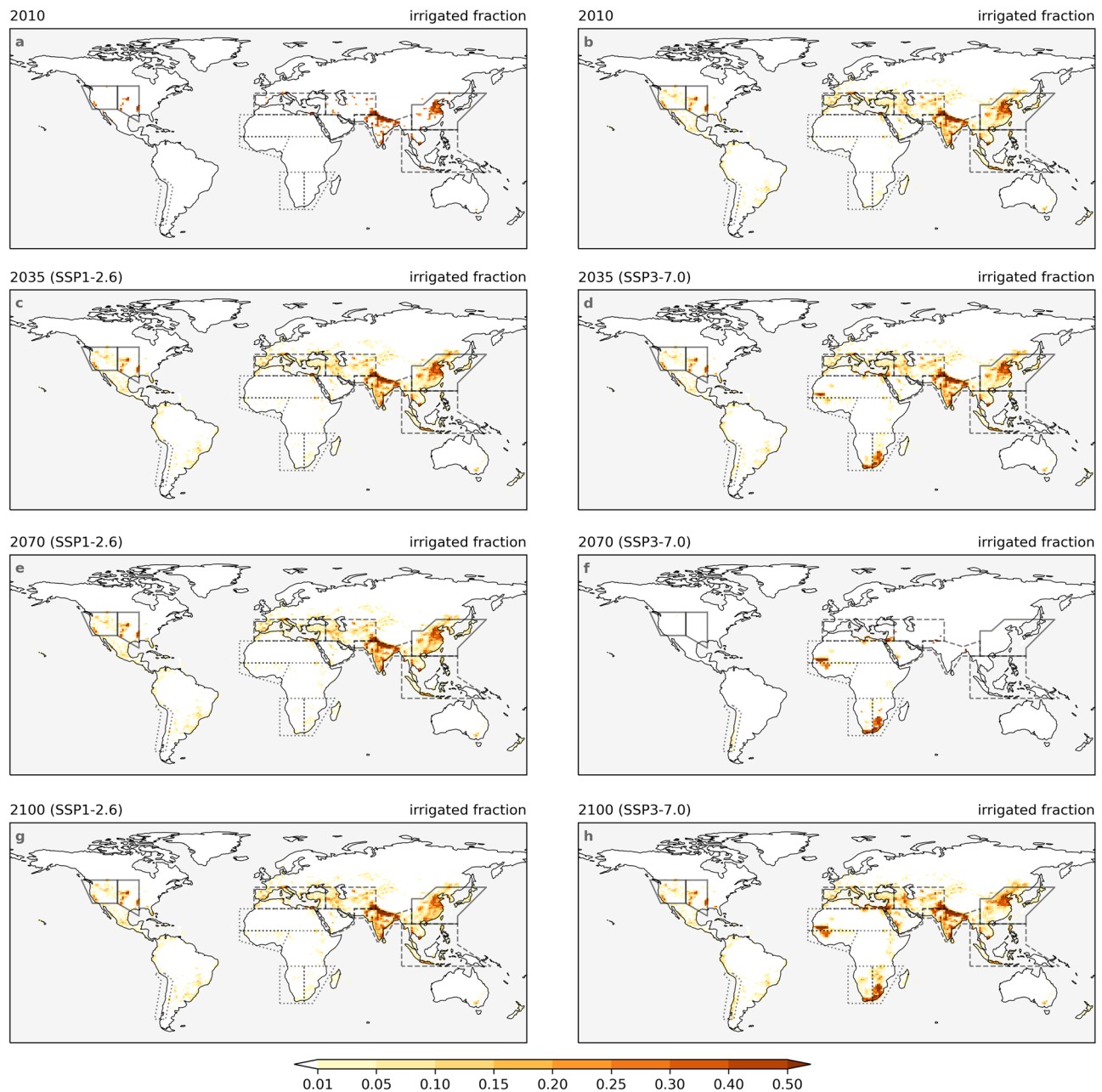

**Fig. 2 | Divergent changes in area equipped for irrigation under two scenarios.** Irrigated fraction of grid cell in the years 2010 (**a**, **b**), 2035 (**c**, **d**), 2070 (**e**, **f**), and 2100 (**g**, **h**), under SSP1-2.6 (**left column**) and SSP3-7.0 (**right column**). The spatial coverage of the IPCC reference regions[32] used in this study is indicated by solid lines, dashed lines, and dash-dotted lines.

3-hour periods when the wet-bulb temperature ($T_w$) exceeds the 99th percentile during the period 1985-2014 in the ensemble without irrigation (Fig. 5b). The regions experiencing the highest dry-heat temperatures also often face extreme moist-heat temperatures, such as North India and the western part of the Sahara, with a 99th percentile value of $T_w$ exceeding 26 or even 28 °C.

Historical irrigation has cooling impacts on dry-heat extremes (Fig. 5c), reducing the exposure hours of this event in many areas in South Asia, East Asia, West Central Asia, and Central North America, by more than 25 hours $yr^{-1}$ (28%). Unlike its impacts on dry-heat extremes, historical irrigation slightly increases the frequency of moist-heat extremes, particularly in specific regions such as West Central Asia, the Mediterranean, and Southeast Asia, where their annual occurrence has increased by more than 25 hours (Fig. 5d). The cooling effects on

dry-heat extremes and the warming effects on moist-heat extremes are both generally projected to amplify in future periods under SSP3-7.0. For instance, the annual exposure to dry-heat in certain grid cells in South Asia is projected to decline by ≥100 hours $yr^{-1}$ due to irrigation (Fig. 5g). Likewise, in some grid cells in West Central Asia, the irrigation-induced increase in annual moist-heat exposure also surpasses 100 hours $yr^{-1}$ (Fig. 5h). However, under SSP1-2.6, irrigation-induced impacts are expected to vary across regions. In areas with higher socio-economic capacity, such as Central and West North America, the effects of irrigation on dry-heat extremes nearly vanish, whereas in lower-capacity regions such as South Asia, a comparable enhancement in cooling appears (Fig. 5e). Similarly, the irrigation-induced increase in moist-heat extremes persists and intensifies in

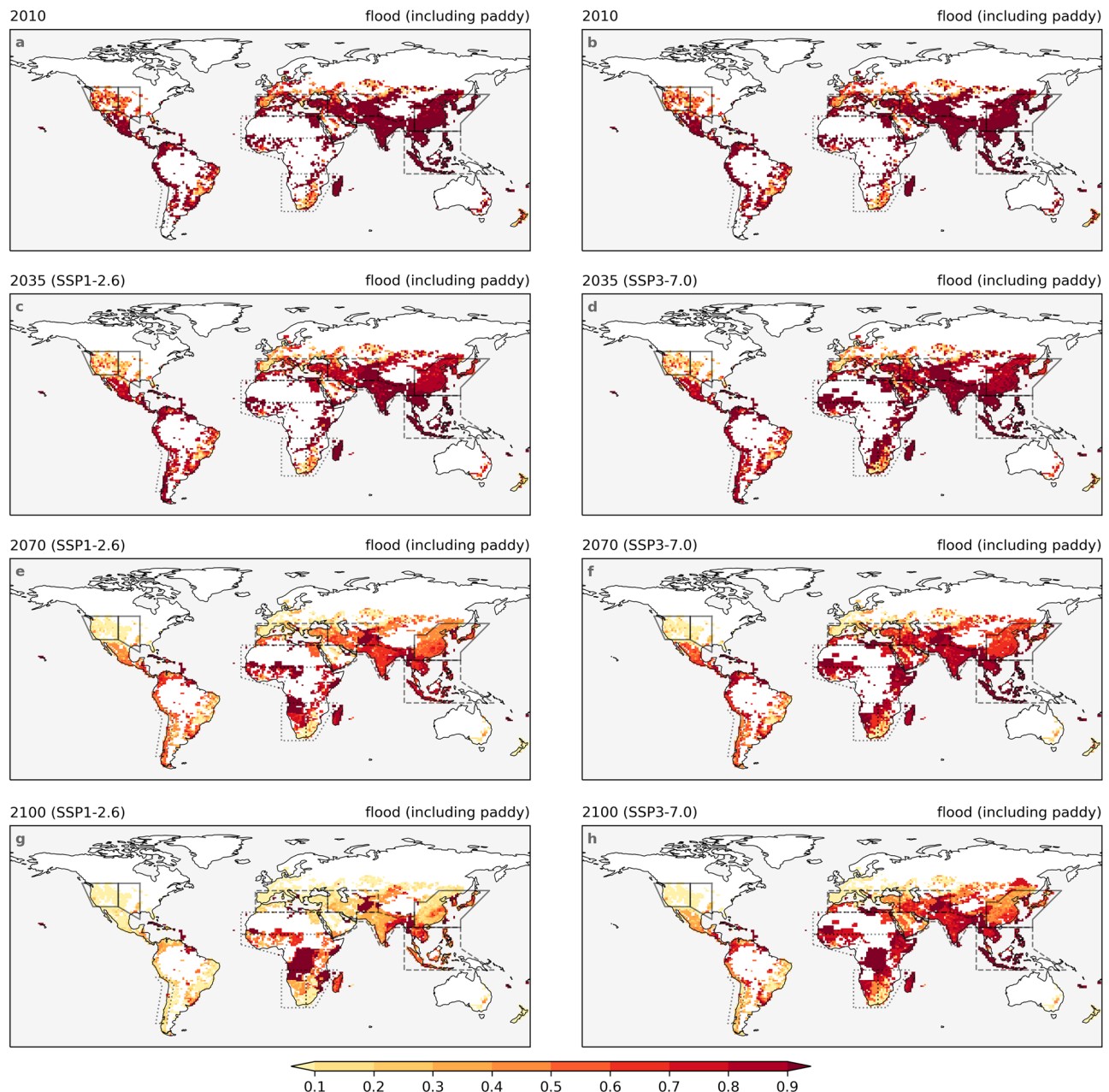

**Fig. 3 | Varying decreasing speed of flood irrigation fraction under two scenarios.** Fraction of flood irrigation in the years 2010 (**a**, **b**), 2035 (**c**, **d**), 2070 (**e**, **f**), and 2100 (**g**, **h**), under SSP1-2.6 (**left column**) and SSP3-7.0 (**right column**). The spatial coverage of the IPCC reference regions[32] used in this study is indicated by solid lines, dashed lines, and dash-dotted lines.

West Central Asia, whereas it weakens in certain grid cells over East Asia (Fig. 5f).

### Rapid increase in exposure to heat extremes caused by climate change

We compare the mean annual exposure hours of extreme dry- and moist-heat events between the periods 2045–2074 and 1985–2014 to quantify changes caused by all forcings (e.g., greenhouse gas emissions change, land-use change, AEI change, and ITS change) under two scenarios. Relative to changes induced solely by irrigation, the combined forcings significantly increase the annual exposure to dry-heat extremes, particularly under SSP3-7.0 (Fig. 6a, c). For instance, in some grid cells across the Arabian Peninsula and East Africa, annual dry-heat extreme hours increase by ≥200 hours (i.e., ≥2.2 times more likely) under SSP1-2.6 and by ≥400 hours (i.e., ≥4.5 times more likely) under

SSP3-7.0, despite the slight reduction in local dry-heat hours due to irrigation changes (Fig. 5c, e, g). Even in South Asia, where irrigation-induced cooling is most pronounced, dry-heat exposure still increases by more than 50 and 100 hours under the two scenarios, respectively. These findings suggest that, despite increased cooling impacts, irrigation alone cannot counteract the warming trend driven by other forcings. The spatial distribution of differences between the scenarios is relatively uniform, ranging from 50 to 400 hours across the majority of grid cells (Fig. 6e).

The increase in moist-heat extremes driven by all forcings is more substantial than that observed for dry-heat extremes, particularly in low-latitude regions (Fig. 6b, d). For instance, in Central Africa, the annual exposure to moist-heat extremes increases by more than 800 hours (≥8.8 times more likely) under SSP1-2.6, and exceeds 1600 hours (≥17.8 times more likely) under SSP3-7.0. In contrast, in mid- and

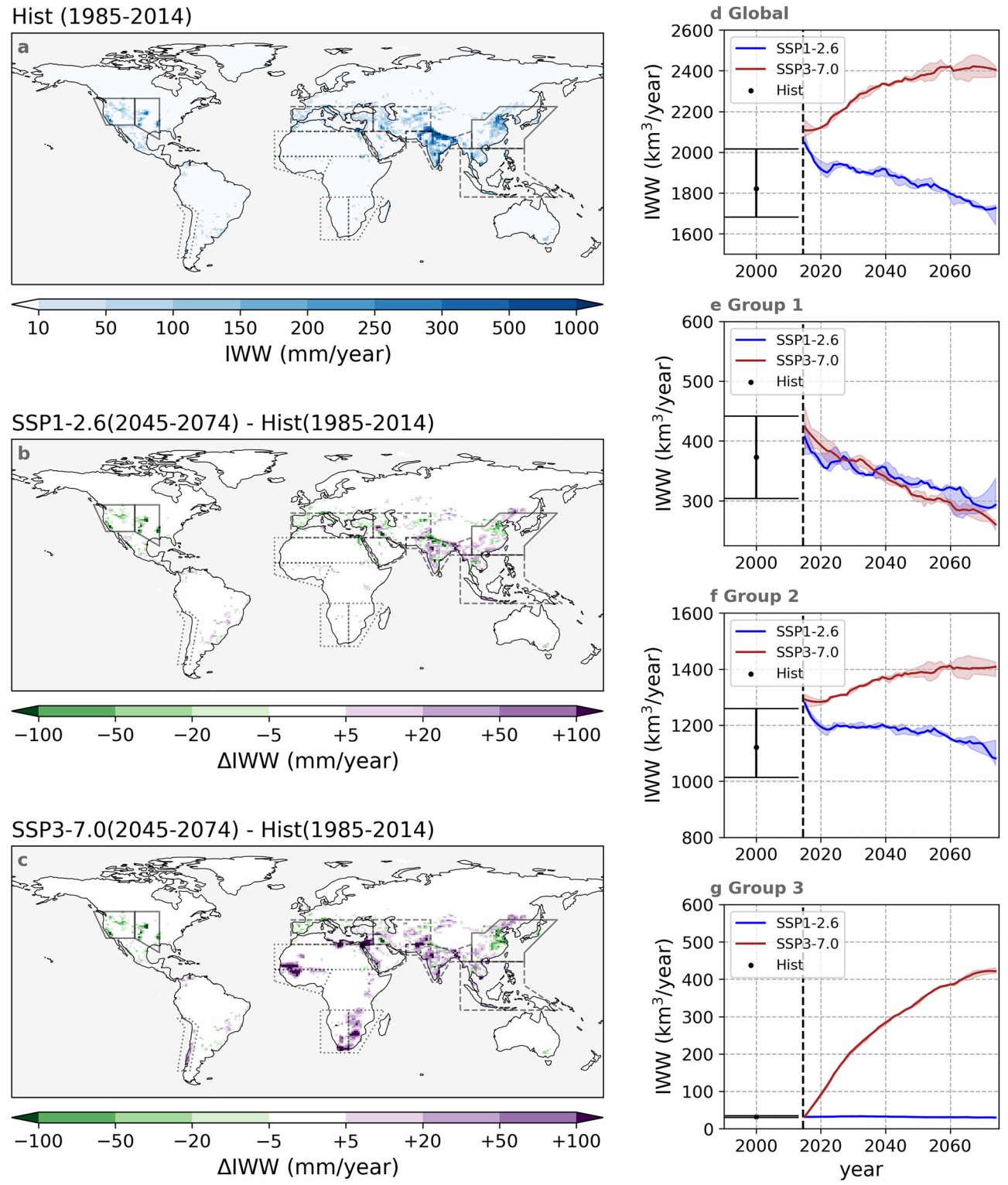

**Fig. 4 | Historical irrigation water withdrawal and projected changes under SSP1-2.6 and SSP3-7.0 scenarios.** Annual mean irrigation water withdrawal (IWW) in the historical period (1985-2014, **a**). Projected changes in annual mean IWW in future period (2045-2074) under SSP1-2.6 (**b**) and SSP3-7.0 (**c**), compared to the historical period. The values shown here are the mean values of all three ensemble members. The spatial coverage of the IPCC reference regions[32] used in this study is indicated by solid lines, dashed lines, and dash-dotted lines. Global (**d**) and sum of regional annual IWW of three groups of regions (**e–g**) during the historical (1985–2014) and future (2015–2074) periods under SSP1-2.6 and SSP3-7.0.

Historical IWW is visualised as a single bar in the year 2000, as in the historical simulations, AEI and ITS are static. The bar indicates the mean value and the range indicates the maximum and minimum values of the 30-year period. In future periods, ranges indicate the maximum and minimum values simulated by three ensemble members and lines indicate the median values. All values in the future period have been smoothed using Savitzky-Golay filtering (order = 2, window = 7)[71]. Regions of each group are listed in Table S1, and their locations are visualised in panel **a**. IWW of individual regions are visualised in Fig. S6.

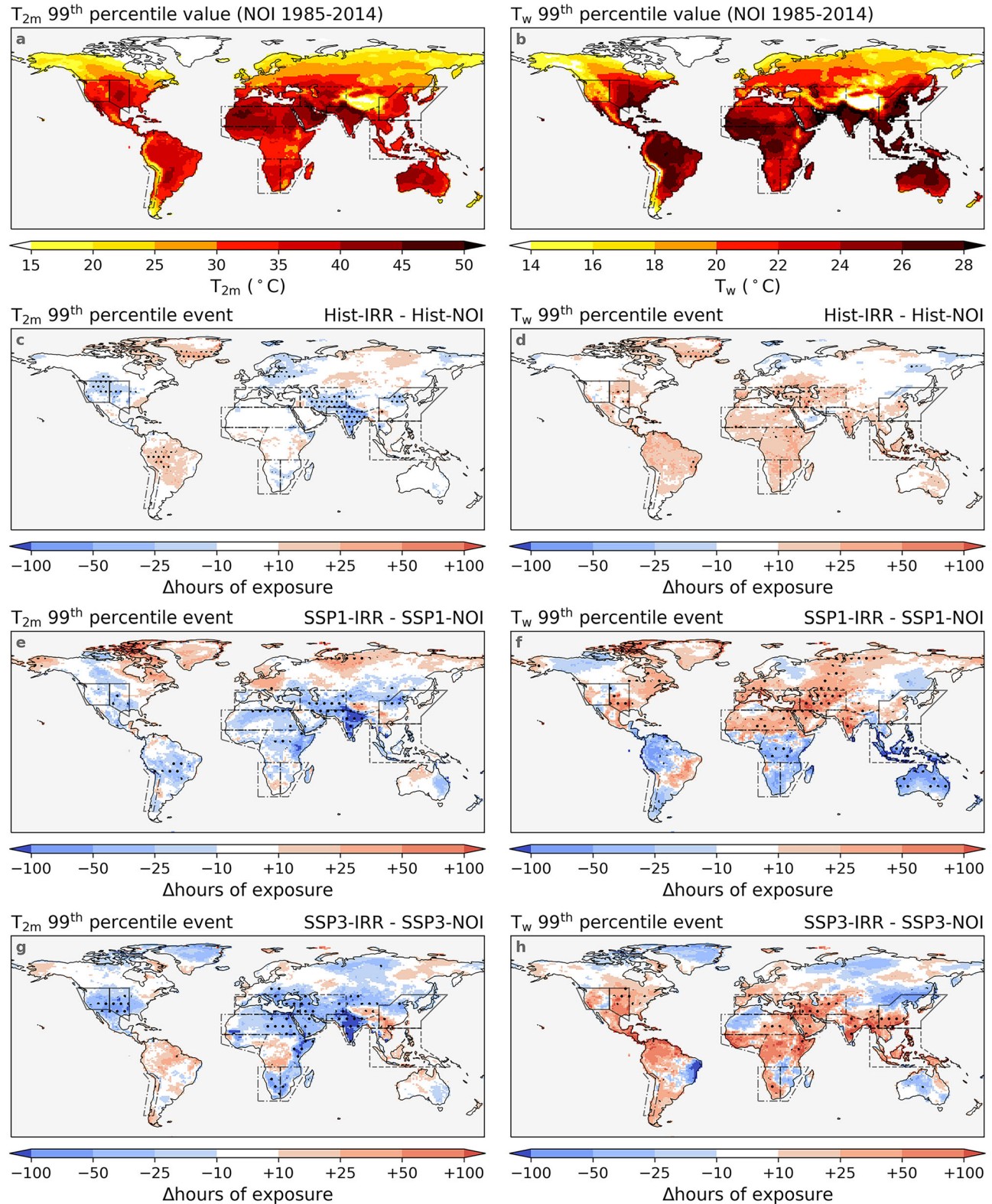

**Fig. 5 | Dry and moist-heat extremes and impacts of irrigation on them.** The 99th percentile values of 2-metre air temperature ($T_{2m}$: **a**) and wet-bulb temperature ($T_w$: **b**) of the historical simulations (1985–2014) without irrigation (Hist_NOI). Annual hours changed by historical irrigation of exposure to extreme events when ($T_{2m}$: **c** and $T_w$: **d**) exceed the values shown in **a**, **b**. The calculation of impacts of forcings is conducted by subtracting the annual hours of the same events in the historical simulation with irrigation (Hist_IRR) by that of (Hist_NOI), as shown in Fig. 8. **e**, **f** Annual hours changed by future irrigation of exposure to extreme events ($T_{2m}$: **e**, **g** and $T_w$: **f**, **h**) under SSP1-2.6 (**e**, **f**) and SSP3-7.0 (**g**, **h**). The calculation of

impacts of forcings is conducted by subtracting the annual hours of the same events in the future simulation with irrigations (SSP1_IRR and SSP3_IRR) from those of future simulations without irrigation (SSP1_NOI and SSP3_NOI) during the period 2045-2074, as shown in Fig. 8. The values shown here are the mean values across all three ensemble members. Hatches indicate that all three ensemble members agree on the direction of change ($\leq$-10 or $\geq$+10 hours yr$^{-1}$). The spatial coverage of the IPCC reference regions[32] used in this study is indicated by solid lines, dashed lines, and dash-dotted lines.

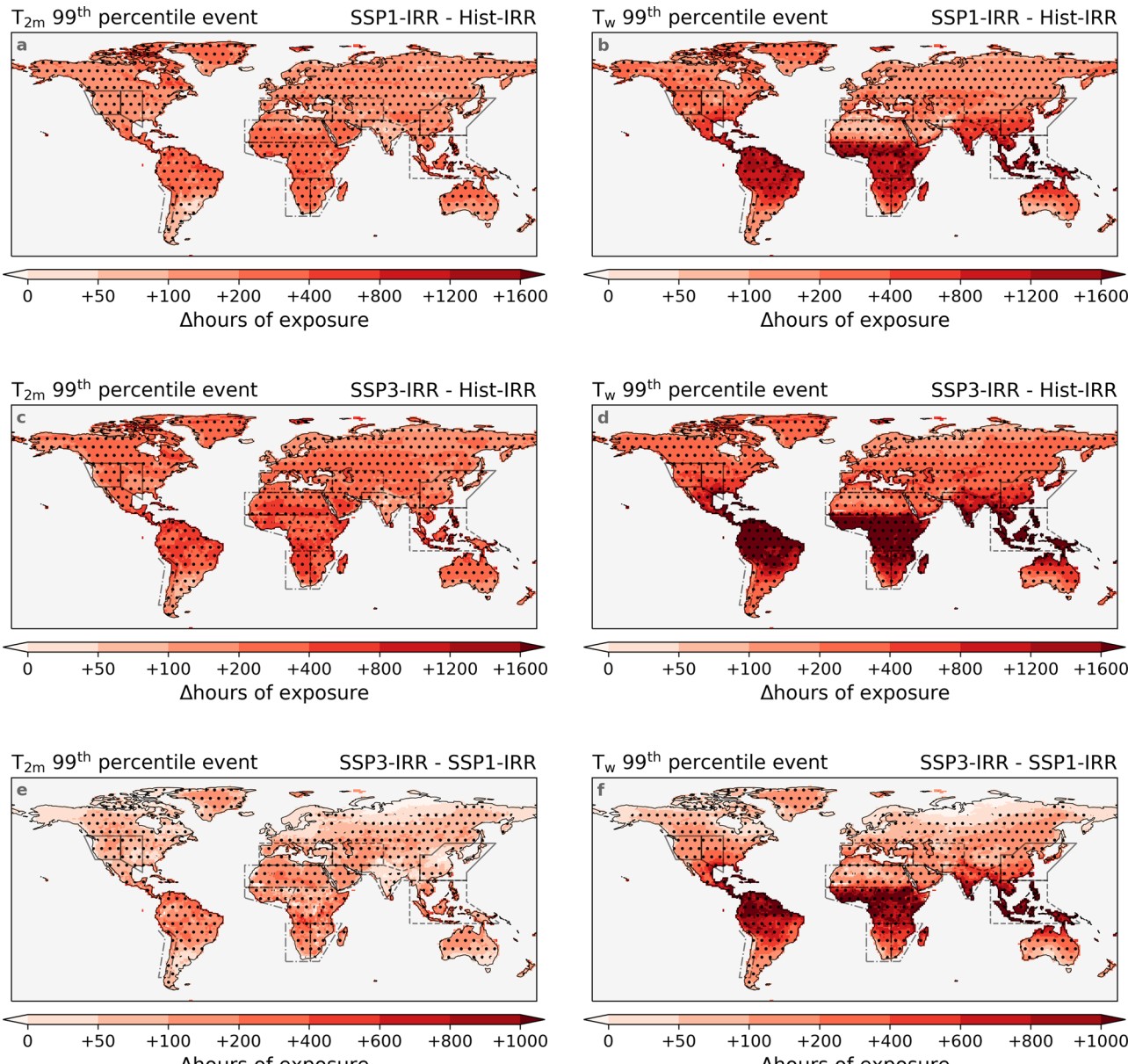

**Fig. 6 | Substantially increased extreme exposure due to climate change.**
Impacts of all forcings (greenhouse gas emissions change, land-use change, irrigation change, etc.) on the annual hours exposed to dry- (**a**, **c**) and moist-heat extremes (**b**, **d**) under SSP1-2.6 (**a**, **b** and SSP3-7.0 (**c**, **d**). The extremes are defined as the exceedance of the 99th percentile values of 2-metre air temperature ($T_{2m}$) and wet-bulb temperature ($T_w$) of the historical simulations (1985–2014) without irrigation (Hist_NOI), as shown in Fig. 5a, b. The calculation of impacts of forcings is conducted by subtracting the annual hours of the same events in the historical simulation with irrigation (Hist_IRR: 1985–2014) from those of future simulations with irrigation (SSP1_IRR and SSP3_IRR: 2045-2074), as shown in Fig. 8. The difference in annual hours exposed to dry- (**e**) and moist-heat extremes (**f**) under two scenarios (calculated by subtracting the annual mean hours in SSP1_IRR from SSP3_IRR). The values shown here are the mean values across all three ensemble members. Hatches indicate that all three ensemble members agree on the direction of change (≤-10 or ≥+10 hours yr⁻¹). The spatial coverage of the IPCC reference regions[32] used in this study is indicated by solid lines, dashed lines, and dash-dotted lines.

high-latitude regions such as western North America, this increase is limited to ≤400 and ≤800 hours, respectively. The spatial pattern of scenario differences is similar, with increases exceeding 800 hours in tropical regions and ranging from 50 to 200 hours across most grid cells at higher latitudes (Fig. 6f). The contribution of irrigation to moist-heat exposure is relatively modest, with a few notable exceptions in arid regions characterised by high IWW. For example, in West Central Asia, where total forcings lead to increases of 200–400 hours and 400–800 hours, irrigation alone contributes an increase of ≥100 hours in certain grid cells under both scenarios (Fig. 5f, h). The results suggest that, under equivalent changes in greenhouse gas

emissions and land use, the frequency of moist-heat events is more substantially impacted than that of dry-heat events. Consequently, great emphasis should be placed on assessing and mitigating the risks associated with moist-heat exposure in a warmer world.

## Difference between dry- and moist-heat extremes hours increase enlarged by irrigation

Given the strong correlation between AEI and population density[5], we analyse the annual average hours of dry- and moist-heat extremes over intensively irrigated regions as a proxy for human exposure to extreme events. To this end, we consider two land categories, distinct from the

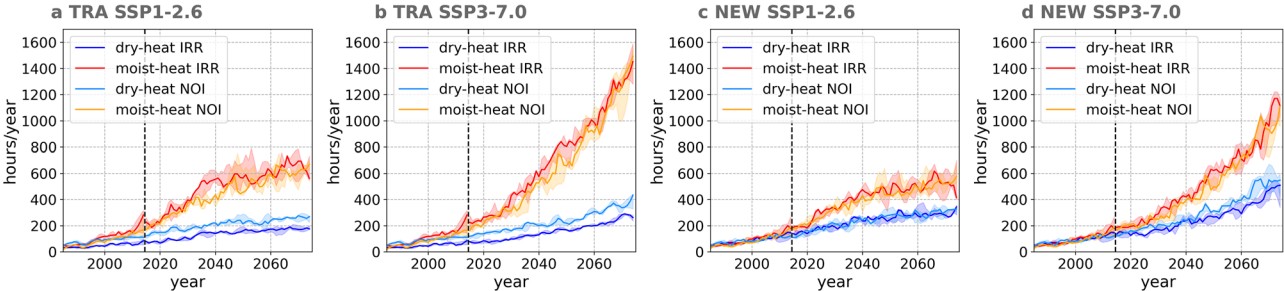

**Fig. 7 | Irrigation enlarges the difference between dry- and moist-heat extremes exposure.** Average annual hours of exposure to dry- and moist-heat extreme event over traditional hot spot grid cells (Fig. S7a) under SSP1-2.6 (**a**) and SSP3-7.0 (**b**). Average annual hours of exposure to dry- and moist-heat extreme events over traditional hot spot grid cells (Fig. S7b) under SSP1-2.6 (**c**) and SSP3-7.0 (**d**). The vertical dashed line indicates the year 2014, which is the ending year of the historical period. Ranges indicate the maximum and minimum values simulated by three ensemble members, and lines indicate the median values. The time series has been smoothed using Savitzky-Golay filtering (order = 2, window = 7)[71].

three groups previously introduced, namely traditional irrigation hotspot areas, defined as grid cells where more than 20% of the area was irrigated during the historical period, and new irrigation hotspot areas, defined as grid cells where irrigation coverage was below 20% historically but exceeds 20% by the year 2070 (Fig. S7). Both types of extreme events become more frequent in the future across both scenarios, with moist-heat exposure hours increasing more rapidly than those for dry-heat, as indicated by the spatial patterns (Fig. 7). For instance, under SSP3-7.0, mean dry-heat exposure in traditional hot spot areas rises from under 100 hours during the historical period to approximately 250 hours in the 2070s, while moist-heat exposure increases from 300 hours to around 1400 hours (Fig. 7b, d). The divergence between scenarios becomes apparent after 2035, beyond which the rate of increase in exposure slows under SSP1-2.6.

Over traditional hotspot areas, irrigation-induced cooling effects on dry-heat and concurrent warming effects on moist-heat contribute to an amplified divergence in exposure hours between these two types of extreme events. For instance, under SSP3-7.0, annual exposure to dry-heat extremes is reduced by 50–100 hours, while exposure to moist-heat extremes increases by 20–100 hours (Fig. 7b). A comparable pattern is evident over new irrigation hotspot areas, where the difference in exposure between the two extremes can also widen by up to 100 hours (Fig. 7d). These findings underscore again the importance of monitoring local moist-heat metrics, as increases in moist-heat exposure do not follow a simple linear relationship with changes in dry-heat exposure. Furthermore, over intensely irrigated areas, irrigation plays an important role as an anthropogenic climate forcing, which should not be neglected in future studies.

## Discussion

The assumptions regarding land use and agriculture under SSP1 and SSP3 are provided as "SSP1, a green road; strong regulations to avoid environmental tradeoffs; improvements in agriculture productivity; rapid diffusion of best practices", and "SSP3, a rocky road; hardly any regulation; continued deforestation due to competition over land and rapid expansion of agriculture; Low technology development, restricted trade"[33]. The LUH2 dataset[31] agrees with those narratives, with the key feature that AEI keeps almost constant under SSP1-2.6 and expands rapidly under SSP3-7.0 (Figs. 1, 2). Some assumptions regarding the irrigation water use efficiency have also been proposed, with "large improvements in irrigation water use efficiency where possible" for SSP1 and "only modest improvements in irrigation water use efficiency" for SSP3[25]. The generated ITS data align well with those assumptions, with a rapid decrease in traditional inefficient flood irrigation under SSP1-2.6 and a slower decrease under SSP3-7.0 (Figs. 1, 3). Although we apply a simplified adjustment factor to account for annual ITS changes based on socio-economic and hydro-climatic

variables (see Methods), the resulting estimates are consistent with prevailing assumptions, thereby offering a valuable enhancement to existing datasets.

At the global scale, projected IWW shows opposite trends under SSP1-2.6 and SSP3-7.0 (Fig. 4d), but the trend varies regionally (Fig. 4e–g, S6). Under SSP1-2.6, IWW is projected to decline in many regions, even without any contraction in irrigation extent. This finding highlights the potential of a transition to efficient irrigation techniques as a method to safeguard future water resources, corroborating a previous study based on idealised scenarios[27]. Oppositely, under SSP3-7.0, IWW is projected to steadily increase over socio-economically less-developed regions following irrigation expansion – both in traditional hot spot regions (Fig. 4f, S6d–h) and in emerging irrigation hot spot regions in Africa (Fig. 4g, S6i–m). This irrigation expansion is largely triggered by an increasing demand for food caused by local population growth[34,35]. However, many of these regions have already been classified as having unsustainable irrigation or being unsuited for sustainable irrigation expansion[36,37]. In this study, water availability is not activated, so the simulated IWW mostly indicates irrigation demand, and the requirement for water may therefore not be fulfilled, leaving crop yields in danger. More troublesome, the growth in population will also increase the water demand from other sectors such as domestic, livestock, and industry, which will further aggravate local competition for water[38].

Limiting unsustainable irrigation expansion and enhancing irrigation efficiency are, therefore, crucial for mitigating water scarcity. Achieving this requires a thorough assessment of current and projected hydrological conditions prior to selecting areas for cropland expansion, as well as the development and deployment of high-efficiency irrigation technologies, particularly in regions already facing water stress. In addition to these fundamental measures, several supplementary strategies may help delay the onset of regional and global water shortages, like optimising the global crop trade network to achieve virtual water savings[39], and switching to lower water-consuming diets, as different crops, dairy, and meat products require various amounts of water[40–42]. Both approaches depend on a highly cooperative international framework and heightened public awareness of environmental challenges.

Under both SSP1-2.6 and SSP3-7.0, increased greenhouse gas emissions increase the frequency of extreme dry-heat events (Fig. 6a, c). Irrigation has been proposed as a climate-effective land management strategy[12,13], and has been found to slow or even reverse local warming trends in recent decades[4–6]. However, although the cooling impacts of irrigation on extreme dry-heat persist in projections (Fig. 5e, g), we find that irrigation will no longer be able to create cooling islands in the future as effectively as it did in the past. This could be attributed to the pronounced impacts of climate

change, as well as the upgrade in irrigation techniques over some traditional irrigation regions, as less water will be applied to fields. Notably, the effects of irrigation on dry-heat extreme frequency will be much smaller compared to the impact of following different SSP-RCPs (Fig. 5e, g and 6e), in terms of both magnitude and spatial coverage.

Taking moisture into account dampens or even reverses the beneficial impact of irrigation on thermal comfort in the future. Both traditional hot spots and new hot spots (Fig. 7) will experience more moist-heat extreme events due to irrigation, especially under SSP3-7.0. Many irrigation hot spot regions, like East China, North India, and West Africa, are projected to become the most densely populated regions in the future[43], further contributing to the overall risk of irrigation-induced humid-heat extremes. We find that the frequency of moist-heat extremes is more sensitive to the SSP-RCPs than the frequency of dry-heat extremes, which highlights again the benefits of the sustainable development of irrigation. As efficient irrigation techniques reduce water use, they could be a solution to decrease irrigation-induced moist-heat stress[22]. Overall, this study reveals the increasing risk of water scarcity, dry- and moist-heat extremes under future climate change and irrigation scenarios, which highlights the importance of limiting greenhouse gas emissions and improving irrigation efficiency.

In this study, we estimate the changes in irrigation techniques under socioeconomic and greenhouse gas concentration scenarios, drawing on the differences in socioeconomic development stage and baseline aridity level of every country (see Methods). However, uncertainties remain in this dataset because of both the lack of observational data and the assumptions applied. The baseline global gridded dataset for present-day irrigation techniques distribution[27] is not a directly observed dataset. Instead, it was generated using a decision tree algorithm based on country-level data from multiple sources[44–46], further informed by global cropland maps. Moreover, in this baseline dataset, flood irrigation and paddy irrigation are not distinguished as separate categories. Instead, paddy irrigation is included under the broader classification of flood irrigation, despite the practical differences between the two methods in actual implementation. In the future, more information should be collected through the national agricultural census, and the classification methods to distinguish different irrigation techniques based on remote sensing data need to be developed. This will provide a more realistic basis for projecting the distribution of irrigation techniques and will facilitate the understanding of key factors for improving irrigation efficiency.

Regarding the assumptions applied for transitions in ITS, only socioeconomic and hydro-climatic conditions are considered as drivers. In reality, farmers' individual decision-making and government policies are not fully determined by socio-economic capacities. For example, farmers' decisions to improve irrigation efficiency are motivated by multiple factors[47], especially the cost and benefits[48]. Another issue is that the socio-economic capacity variables are assumed to be the same within each country, ignoring the internal variability within larger countries. Moreover, a 1% default rate of flood irrigation reduction is assumed for all countries based on historical time series of two countries (see Method), which may not be realistic for other countries, and changing this default rate may lead to different results. Furthermore, precipitation is the only variable used to indicate hydro-climatic complexity, but some other variables, like groundwater availability, river discharge, should also be included if data availability permits. At the same time, the feedback of socio-economic capacities to irrigation techniques shares changes, which should also be considered in future work. This will support the development of more realistic datasets on the share of irrigation techniques consistent with future pathways, enabling more accurate predictions of the changes in irrigation practice.

The CESM2[49] was recently expanded to represent various irrigation techniques[19], and we here extend this functionality to capture transient changes in these techniques. Although the implementation of multiple irrigation techniques has led to improved performance and utility[19], limitations remain in the model's crop and irrigation representation. For example, the CESM2 has a globally identical crop and irrigation technique parameterisation for each crop type, and only one single cropping season is allowed per year. Furthermore, in this study, the limitation of water withdrawal for irrigation is not activated, as there is currently no implementation of groundwater availability. To further improve irrigation representation in Earth system models, new research could aim at collecting and calibrating crop- and irrigation-related parameters regionally, implementing crop rotations and multiple cropping seasons, and incorporating groundwater availability and evolution. At the same time, the increased complexity of the model in this study (cropland columns expanded from 64 to 128) requires more computational resources, underscoring the importance of evaluating the new module on reproducing surface fluxes. Previous evaluations were mainly focused on a single-point scale[19], and in the future, the validation at regional and local scales should be conducted. These efforts will enable accurate simulation of irrigation-induced impacts on the Earth system while minimizing unnecessary consumption of computational resources.

Generally, the simulation results exhibit a strong dependence on the selected scenarios, with this study focusing exclusively on SSP1 and SSP3, two contrasting extremes in terms of climate mitigation and adaptation risks. However, they may no longer reflect the most current socioeconomic and climate projections, as newer scenarios have since been developed to address contemporary research questions[50]. Future irrigation-related studies should therefore incorporate updated scenario frameworks to yield more relevant insights for policy and planning.

In this study, we, for the first time, design ITS scenarios aligned with the SSP-RCPs, improve an Earth system model to incorporate this information, and perform fully-coupled climate simulations under varying climate and irrigation transition pathways. We then analyse the simulation outputs for irrigation water demand and irrigation-induced impacts on dry and moist-heat stress. The results exhibit a strong sensitivity to the choice of SSP-RCP scenarios, highlighting their important implications for future irrigation planning and development. While these scenarios rely on several idealised assumptions, the underlying narratives still offer valuable insights for guiding future irrigation strategies. More specifically, under SSP1-2.6, conservative irrigation expansion combined with rapid adoption of improved techniques can reduce irrigation water withdrawals, whereas under SSP3-7.0, massive expansion coupled with slow technological upgrades substantially increases withdrawals, generating new irrigation hotspot regions, particularly in Africa. As a scenario with higher greenhouse gas concentrations, SSP3-7.0 results in more frequent dry- and moist-heat extremes than SSP1-2.6, with the increase in moist-heat exposure being more pronounced. Irrigation–particularly in intensely irrigated areas–mitigates dry-heat extremes but slightly amplifies moist-heat, emerging as an important climate forcing that should not be overlooked in future studies.

## Methods

### Projected irrigation techniques share

In this study, we develop a new dataset of Irrigation Techniques Share (ITS) by employing a hydro-economic (HE) conceptual framework[30], which classifies countries based on two variables representing economic and hydrological conditions to project the future ITS under different scenarios. A central assumption of this framework is that countries with high socio-economic conditions and severe hydro-climatic conditions will invest more in high-efficiency irrigation. To verify this assumption and choose the variables representing these

conditions, we calculate Spearman's correlation coefficient between the historical ITS (drip, sprinkler, and flood) at national levels[27] and several variables from the Shared Socioeconomic Pathways (SSPs) datasets[51] and simulation outputs from the Inter-Sectoral Impact Model Intercomparison Project phase 3 (ISIMIP3)[52].

The socio-institutional capacity and hydro-climatic complexity of each country or region derived from those datasets are used to determine future change speed in irrigation system efficiency. SSP narratives were developed to characterize the societal future considering the challenges to mitigation and adaptation[33], independent of greenhouse gas emission scenarios. The SSP variables we use in this study include Gross Domestic Product per capita (GDP)[53], governance strength (GOV)[54], urbanisation (URB)[55], and the Gender Inequality Index (GII)[56]. These datasets each cover more than 170 countries, at a 5-year frequency, from 1985 till 2100. Given the data availability of the hydro-climatic data, we chose SSP1 (the green road: low in both adaptation and mitigation challenges) and SSP3 (the rocky road: high in both challenges) to generate the data.

ISIMIP3 provides a framework under which researchers simulate the responses in different sectors to climate and socioeconomic changes in both historical periods (1850–2014) and future periods (2015–2100) under three SSP-RCP scenarios (SSP1-2.6 and SSP3-7.0), where RCP stands for Representative Concentration Pathway, indicating the level of future greenhouse gas concentrations. In the global water sector, three Global Hydrological Models (GHMs: CWatM[57], H08[58], and WaterGAP2-2e[59]) were forced by meteorological variables from five Earth system models (ESMs): GFDL-ESM4[60], IPSL-CM6A-LR[61], MPI-ESM1-2-HR[62], MRI-ESM2-0[63], and UKESM1-0-LL[64], to simulate the global water cycle at a spatial resolution of 0.5 by 0.5 degrees. We consider three variables: precipitation (P), precipitation divided by potential evaporation (PPET), and terrestrial water storage (TWS), either from the input or output datasets. To represent long-term hydro-climatic conditions, we calculate the 20-year average value (e.g., average value during 1986–2005 to represent the condition of the year 2005), and then average the values of all grid cells per country.

As 2010 is the last year before the projections in SSP dataset, we assume that the data of historical ITS represent the values in the year 2010, although the dataset is from multiple sources across a long period. Thus, we select the socioeconomic data in 2010 and the hydro-climatic data during the period 1996–2010 to perform the calculations. Based on the correlation results (Table S2), we select GDP, GOV, URB and GII to represent socio-institutional capacity and P to represent hydro-climatic complexity. Given that the four socioeconomic variables have high correlations, we conduct a Principal Component Analysis for them and the resulting first four principal components explain 78.2%, 12.0%, 6.4%, and 3.4% of the variance, respectively. In the following analysis, we select the first principal component to represent socio-economic capacities. A high Spearman's rank coefficient indicates a high correlation between two variables, so we assume that a pathway with higher socio-economic capacities tends to have higher motivation to improve irrigation efficiency. Similarly, P also shows a high, though less strong and significant, correlation with historical ITS, and we choose P as the variable representing hydro-climatic conditions (Table S2).

Based on limited and scarce national time series irrigation techniques fraction data from the USA[65] and Iran[66] (Table S3), we assign a default irrigation system update at 1% per year (that is, 1% reduction in the fraction of flood irrigation and 1% increase in the fraction of drip plus sprinkler irrigation). Flood irrigation is replaced by drip and sprinkler irrigation based on the fraction of drip and sprinkler crop functional types (CFTs) (Table S4) in that grid cell. For example, if there are 30 % drip CFTs, 50% sprinkler CFTs and 20 % flood CFTs in one pixel, then a 1% decrease in flood irrigation would cause a 0.375 % (i.e, 30/(30+50)) increase in drip irrigation and another 0.625 % (i.e., 50/(30 + 50)) increase in sprinkler irrigation by default. The fraction is also

limited by the suitability between irrigation techniques of CFTs, because drip irrigation can only be applied over drip CFTs, and sprinkler irrigation can only be applied over drip and sprinkler CFTs (Table S4, as defined before ref. [19]). In other words, the maximum drip irrigation fraction is the fraction of drip CFTs, the maximum sprinkler irrigation is the fraction of drip CFTs plus sprinkler CFTs, and the minimum flood irrigation fraction is the fraction of flood CFTs. We normalise both variables based on the historical data (2010 for the socioeconomic variable and 1996–2010 for the hydroclimatic variable), then classify all countries based on their values. Since the socioeconomic variables consistently have a higher correlation with the ITS relative to the hydro-climatic variables (Table S2), we assign the socioeconomic principal component variable a higher importance (see Table S5). The rate of change is therefore adjusted every five years based on socio-economic and 20-year mean hydrologic conditions.

Given the complexity of the process, a representative example is presented for illustrative purposes (a flowchart is also given as shown in Fig. S8). In the year 2075, a grid cell is assumed to contain 70% flood irrigation, 20% sprinkler irrigation, and 10% drip irrigation. The crop type distribution is assumed to be 35% flood crops, 50% sprinkler crops, and 15% drip crops (see Table S4). The procedure begins by calculating the normalised 20-year average precipitation for the period 2056-2075, which is assumed to be 0.45. Subsequently, the first principal component of the socio-economic indicators–GDP, GOV, URB, and GII–is computed using the PCA model previously developed from historical SSP datasets, yielding a normalised value of 1.25. The annual updating rate is determined by subtracting the hydro-climatic adjusting factor (0.1) from 1 and adding the socio-economic adjusting factor (0.8) (Table S4), resulting in a rate of 1.7% per year over the subsequent five-year period (until 2080) (Table S5). Accordingly, by 2080, the flood irrigation fraction is reduced by 8.5% (1.7% × 5 years), with the reduction redistributed to sprinkler and drip irrigation in proportions of 6.54% and 1.96%, respectively. Thus, by 2080, the shares of flood, sprinkler, and drip irrigation are adjusted to 61.5%, 26.54%, and 11.96%, respectively.

## CESM2 with the new irrigation module and simulations
The Community Earth System Model version 2 (CESM2) is an open-source Earth system model that has been widely used to detect irrigation-induced impacts[5,11,67]. The irrigation module in CESM2 has been expanded to represent four different irrigation techniques: drip, sprinkler, flood, and paddy (used by default if the crop type is irrigated rice)[19]. These irrigation techniques vary in the activation condition, soil water threshold, and target, where the water is applied, and whether surface water is ponded, showing satisfying results for both global and single-point simulations[19]. However, currently, the cropland land unit is divided into 64 columns, one rainfed and one irrigated for all 32 CFTs, which means that only one irrigation method is assigned to each separate CFT per grid cell. To allow three different irrigation techniques for each CFT per grid cell, we expand the cropland unit to 128 columns: one rainfed, one drip irrigated, one sprinkler irrigated, and one flood irrigated for each CFT. The calculations of the land fluxes are conducted individually over each column before being averaged to the grid cell level. Transient ITS is implemented through the preparation of input land use and management time series, in which we provide the annual information of ITS from the year 2015 to 2100. In this study, we do not limit the water withdrawal for irrigation based on river flow, and if river flow cannot meet the water requirement of irrigation, the deficit is extracted from the ocean to close the water balance.

After the model source code development, we create the land use data with irrigation techniques incorporated, based on the land use of the year 2010. In a previous study[27], a static grid cell-scale ITS dataset was developed for the historical period (sources ranging from 1990 to 2010), which is used as the starting point for data generation. The general process is firstly assigning drip irrigation to drip crops (as defined before ref. [19]), as drip irrigation is only suitable for drip crops

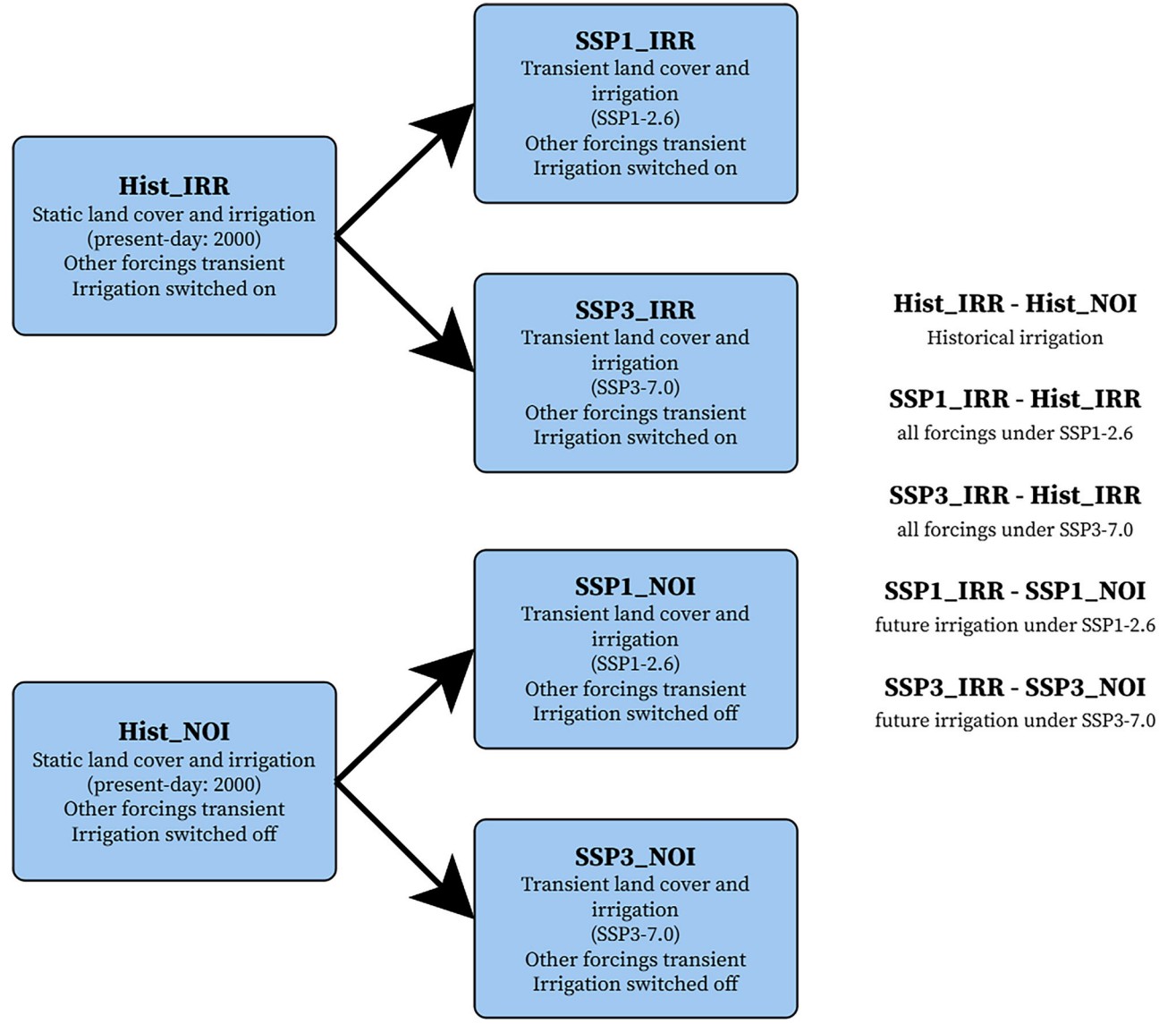

**Fig. 8 | Simulations conducted with CESM2 and the approach to separate impacts.** Historical simulations are conducted from the year 1980 to 2014, with the first five years as a spin-up period. Future simulations restart from historical simulations, from the year 2015 to 2074. By subtracting the dry- and moist-heat extreme event hours from one simulations from another, we calculate the impacts of different forcings, where Hist_IRR and Hist_NOI indicate the simulations with (IRR) and without irrigation (NOI) during the period 1985–2014, SSP1-2.6_IRR and SSP1-2.6_NOI indicate the simulations with (IRR) and without irrigation (NOI) during the period 2045–2074 under SSP1-2.6, and SSP3-7.0_IRR and SSP3-7.0_NOI indicate the simulations with (IRR) and without irrigation (NOI) during the period 2045–2074 under SSP3-7.0.

(see Table S4). Then, sprinkler irrigation is assigned to all sprinkler crops, and in case all sprinkler crops are irrigated by the sprinkler technique, the extra sprinkler irrigation fraction is used for drip crops. After these two processes, flood irrigation is given to all irrigated cropland left. Then, to build the land use time series from 2015 to 2074 under SSP1-2.6 and SSP3-7.0, we performed a linear interpolation to transfer 5-year irrigation technique share data to annual data, then distribute the change in irrigation technique share to different CFTs as mentioned.

We first conduct the two historical simulations with (Hist_IRR) and without irrigation (Hist_NOI) covering the period 1980-2014, with first five years as the spin-up period, using the land use and land management of the year 2010, and including the existing[27] ITS dataset (Fig. 8). Each simulation contains three ensemble members with a very small difference in the initial atmospheric conditions[11]. Then, we generate four ensembles for the future period (2015–2074), containing three members each and branching from the corresponding HIST

simulations (Fig. 8). The future scenarios ensembles encompass two SSP-RCPs (SSP1-2.6 and SSP3-RCP7.0:) and for each emission scenario we generate an ensemble with (IRR) and without (NOI) transient irrigation extent and technique shares (SSP1_IRR, SSP1_NOI, SSP3_IRR, and SSP3_NOI). All simulations employ a horizontal resolution of 0.9° x 1.25°. In the historical simulation, the land use and irrigation technique share are fixed at the present-day level, and other climate forcings (e.g., greenhouse gas emissions) are transient, while in the future simulations, both the climate and land use forcings are transient as well. In CESM2, when transient land-use is activated, the surface map is updated yearly, and the ITS is incorporated in the surface map, which enables the model to use transient ITS. During the simulation, the heat metrics module[68] is switched on, which means that all heat metrics are calculated regularly at a 3-hourly frequency. We focus on only $T_w$, as most other metrics are a weighted average of $T_w$ and $T_{2m}$. In this module, a simplified calculation method of $T_w$ is applied, as shown in

Equation (1):

$$T_w = T_{2m} \arctan(0.151977\sqrt{\phi + 8.313659})$$
$$+ \arctan(T_{2m} + \phi) - \arctan(\phi - 1.676331) \tag{1}$$
$$+ 0.00391838\phi^{3/2} \arctan(0.023101\phi) - 4.68035$$

where $T_{2m}$ is the 2-metre air temperature in degrees Celsius, and $\phi$ indicates 2-metre air relative humidity in %.

## CESM2 output analysis

Heat extreme indices at 3-hourly temporal resolution are obtained from the outputs of simulations, including $T_{2m}$ and $T_w$, which are calculated as described in the module[68]. We compare the multi-year average number of hours per year with extreme dry or humid heat conditions between different combinations of experiments (IRR and NOI) and periods (historical: 1985-2014 and future: 2045–2074) to separate the impacts of all forcings under each SSP-RCP scenario (changes in climate, land use, and irrigation), all forcings between two SSP-RCP scenarios, and irrigation (Fig. 8). To assess the temporal evolution of irrigation-induced impacts, we also calculate annual frequency of dry- and moist-heat extremes (in hours), on average over traditional hot spot areas and new hot spot areas, which are defined as grid cells where more than 20% of their area is irrigated in the year 2010, and grid cells where less than 20% of their area is irrigated in the year 2010 but AEI increases by more than 20% until the year 2070, respectively (Fig. S7).

## Data availability

The data generated for this study have been deposited in the figshare database with the license CC BY 4.0: https://figshare.com/articles/dataset/Yao_et_al_2025_Nature_Communications_Compounding_Future_Escalation/29402120?file=55641308[69]. The raw outputs of simulations can be obtained by inquiring the corresponding author.

## Code availability

CESM2 is publicly available through the Community Earth System Model (CESM) repository (https://github.com/ESCOMP/CESM.git) and the version 'release-cesm2.2.0' is used in this study. All scripts developed for this study are available at: https://github.com/YiYao1995/Yao_et_al_2025_Nature_Communications_Compounding_Future_Escalation.git[70].

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

## Acknowledgements

The computational resources and services used in this work for the simulations and storage of CESM data were provided by the VSC (Flemish Supercomputer Center), funded by the Research Foundation - Flanders (FWO) and the Flemish Government, department EWI. S.D.H. acknowledges funding by BELSPO (B2/223/P1/DAMOCO). S.L. was supported by a PhD Fundamental Research Grant by Fonds Wetenschappelijk Onderzoek - Vlaanderen (11M7725N). W.T. acknowledges

funding from the European Research Council (ERC) under the European Union's Horizon Framework research and innovation programme (grant agreement No 101124572; ERC Consolidator Grant 'LACRIMA'). J.J. was supported by the NASA GISS Climate Impacts Group, the Future of Life Institute, and the US Department of Agriculture. Y.S. was supported by JSPS KAKENHI (Grant Number 22K14101). The authors acknowledge the DiscussCESM forum community for all the help regarding model source code development and input dataset preparation.

## Author contributions

Y.Y. and W.T. designed the study. Y.Y. performed all analyses under the supervision of W.T. and wrote the manuscript with support from Y.S., N.V.M., S.T., J.K., S.J.D.H., S.L., D.M.L., W.J.S., Y.W., A.D., B.I.C., S.I.S., L.L., J.R.B., J.J., and W.T. Y.Y., Y.S., N.V.M., S.T., S.L., D.M.L., W.J.S., Y.W., J.J., and W.T. designed the irrigation techniques change projections. Y.Y. modified the source code of CESM2 with the help of W.J.S. Y.Y. performed the simulations with the help of S.D.H, D.M.L., and W.J.S. J.R.B. provided insights for interpreting heat stress metrics. A.D., B.I.C., and S.I.S. provided valuable insights and comments on the discussion.

## Funding

## Competing interests

The authors declare no Competing Interests.

## Additional information

[1]Department of Water and Climate, Vrije Universiteit Brussel, Brussels, Belgium. [2]Institute for Atmospheric and Climate Science, Department of Environmental Systems Science, ETH Zurich, Zurich, Switzerland. [3]Japan Agency for Marine-Earth Science and Technology, Yokohama, Japan. [4]Institute for Environmental Studies (IVM), Vrije Universiteit Amsterdam, Amsterdam, Netherlands. [5]Forecast and Services Department, European Center for Medium-Range Weather Forecasts, Bonn, Germany. [6]Q-ForestLab, Department of Environment, Universiteit Gent, Ghent, Belgium. [7]NSF National Center for Atmospheric Research, Boulder, CO, USA. [8]Biological and Environmental Science and Engineering Division, King Abdullah University of Science and Technology (KAUST), Thuwal, Saudi Arabia. [9]Laboratory 7619 METIS, Sorbonne Université, CNRS, EPHE, IPSL, Paris, France. [10]NASA Goddard Institute for Space Studies, New York, NY, USA. [11]Lamont-Doherty Earth Observatory, Palisades, NY, USA. [12]Department of Geography, The University of Hong Kong (HKU), Hong Kong, China. [13]Institute for Climate and Carbon Neutrality, The University of Hong Kong (HKU), Hong Kong, China. [14]Department of Chemistry and Bioscience, Aalborg University, Aalborg, Denmark. [15]Columbia University, Climate School, New York, USA. [16]Potsdam Institute for Climate Impact Research (PIK), Member of the Leibniz Association, Potsdam, Germany. ✉e-mail: yi.yao@vub.be

