## [Transparent Peer Review file · Nature Communications]

Compounding future escalation of emissions- and irrigation-induced increases in humid-heat stress

Corresponding Author: Dr Yi Yao

Version 0:

Reviewer comments:

Reviewer #1

(Remarks to the Author)

Review of "Compounding future escalation of emissions- and irrigation-induced increases in humid-heat stress" submitted to Nature Communications for consideration for publication.

The manuscript describes a global study on the impact of climate change on irrigation development under 2 SSPs. The authors developed a new irrigation module for CESM2, which now considers more irrigation techniques. The model is then applied under a historical scenario and for different SSPs in the 21st century. The results show that irrigation demand will decrease under SSP1 and increase under SSP3. Based on the results of the simulations, the authors also determined the impacts on dry-heat and moist-heat stress. These results show moderate changes for dry-heat stress, but substantial increases for moist-heat stress under the different SSPs. The study seems a valuable addition to the literature, with novel results related to the projection of irrigation for the 21st century, including the impacts on dry-heat and moist-heat stress.

There are two issues regarding the considered SSPs that would need some more explanation in the manuscript. The authors mainly focus on two SSPs, i.e. SSP1 and SSP3, while SSP5 is also considered, but not fully included in the results. This is somewhat confusing. Please provide the reasoning why SSP5 is not fully considered. In either case, it would make sense to either exclude SSP5 from the manuscript all together or better integrate the SSP5 results in the manuscript.

In addition, I was not expecting that SSP3 would result in a such a substantial increase in irrigation. Following O'Neill et al. (2017), it seems that SSP5 is a more likely pathway that would result in such an increase in irrigation, given the assumption that agriculture will be "Highly managed, resource-intensive; rapid increase in productivity" under SSP5. Agriculture under SSP3 is suggested to be characterized by "Low technology development, restricted trade", which does not suggest such a high focus on irrigation, especially drip irrigation. The description of how the different SSPs affect irrigation development fall short in this respect (lines 573-586). Hence, the authors should better explain how the different SSPs are implemented and discuss in the main document how the implementation of the SSPs compare to the literature, e.g. O'Neill et al. (2017).

O'Neill, B.C., Kriegler, E., Ebi, K.L., Kemp-Benedict, E., Riahi, K., Rothman, D.S., van Ruijven, B.J., van Vuuren, D.P., Birkmann, J., Kok, K., Levy, M., Solecki, W., 2017. The roads ahead: Narratives for shared socioeconomic pathways describing world futures in the 21st century. *Global Environmental Change* 42, 169–180.
<https://doi.org/10.1016/j.gloenvcha.2015.01.004>

Considering the results, the authors show the results for dry-heat and moist-heat stress for scenarios with and without irrigation. I am not fully convinced of the added value of showing the results without irrigation, at least in the main document. The first part of the Results section solely focuses on how irrigation is projected to change under the different SSPs. In the second part the authors also consider scenarios without irrigation. Although these scenarios can show how heat stress is affected by climate change only, they are not realistic scenarios, but more "what if" scenarios. I suggest to move the scenarios without irrigation to the Supporting Material and mostly focus on the scenarios with irrigation, also when presenting the results in the text. This would significantly improve the readability of the text, where especially lines 176-222 are sometimes difficult to follow because of the number of different comparisons that are made by the authors, i.e. historical vs SSP and irrigation vs no irrigation between the SSPs and between historical and the SSPs. In that sense, for Figures 4 and 5, I would keep panels a, c and d in the main document, while b, e and f can be moved to the Supporting Material.

The concluding remarks focus mostly on the suggestion to move to an SSP1 world (lines 321-330 and lines 341-353). From the perspective of sustainability, the goal to decrease irrigation water use and the impacts on heat stress, I fully agree with this. But is this also a realistic goal? Since the authors considered only two SSPs, which are in most aspects also their opposites, it seems logic that the authors come to the conclusion that SSP1 would be better aim. However, it is likely that other SSPs may be more realistic given the ongoing socioeconomic and climatic developments. More discussion is needed that would need the involvement of the other SSPs, given their particularities and possible impacts on irrigation and heat stress.

Below I have provided specific comments to the text, figures and tables.

Specific comments

Lines 31-32: With "...its climatic impacts in future with possible changes in both extent and efficiency considered" the authors mean "... its future climatic impacts considering possible changes in both extent and efficiency".

Line 42: Replace "increase" with "increases".

Line 43: The authors mean ≥ 100 hours yr⁻¹? Please revise.

Lines 93-94: Why is it relevant to mention that SSP5-8.5 is also simulated but not considered? Please clarify in the text.

Lines 96-98: The text between the parentheses depends on the socioeconomic scenario, right? Please clarify this in the text.

Lines 110-113: So, the irrigated area is based on LUH2? This has not been mentioned in the Methods section. I suggest to revise this sentence and start the sentence with a result from this study, instead of with the projections from LUH2.

Figure 1: Please revise the figure caption, where it says two times "equipped", while one time seems sufficient.

Figure 2d-2g: It is difficult to understand where the historical IWW results are shown. I suggest to make the grey bar as wide as the historical period (1985-2014), to make this stand out more. Moreover, I don't see the point of the vertical green and red lines for the two SSP scenarios. I suggest to remove these lines and only include the trend line.

Lines 131-151: The results of Figure 2 are presented here, going progressively from panel a to g. I suggest to integrate better the results of panels a-c and panels d-g. For instance, by presenting first the results of SSP1 and discussing simultaneously the results of panels d-g for this scenario, etc.

Lines 282-288: I would say that farmers' individual decision-making and government regulation are part of the socioeconomic conditions. But I can be wrong.

Lines 270-305: Limitations are usually discussed at the end of the Discussion section.

Lines 583-586: Please explain here why the authors focused on these two extreme SSPs, i.e. SSP1 and SSP3 (not considering SSP5). Moreover, in the main text the authors focus on SSP1 and SSP3. It is a bit confusing that results are shown for SSP5 also, but not really considered in the main document. I suggest to remove this from the manuscript or incorporate the results in the main text.

Lines 620-632: I suggest to include a flow chart to make this easier to understand.

Line 671: Please give the definition of NOI.

(Remarks on code availability)

A README file is provided, but only gives limited information on what the different scripts accomplish.

In general, most scripts do not include comments on the functionality of the script. It is therefore difficult to understand what the code is meant for. Moreover, the script are not particularly DRY (Don't Repeat Yourself), there is a lot of repetition, which makes the code difficult to understand. Also the code is written in three different programming languages (Matlab, Python and shell script), which is of course not ideal.

Reviewer #2

(Remarks to the Author)

General Comments:

This is an interesting article that investigates the irrigation impacts on extreme dry and humid heat under future scenarios of changes in irrigation extent and irrigation efficiency. The authors set up different SSP-RCP pathway scenarios to develop dynamic global irrigation techniques share (ITS) datasets and embed them in the CESM2 Model, and conduct sensitive simulation experiments with CESM2 to assess irrigation water withdraw and quantify irrigation impacts on extreme dry and humid heat under dynamic conditions of changing irrigation extent and efficiency. The innovation of this study is the development of the dataset that considers irrigation technology change using a combination of SSP and ISIMIP3 scenarios embedded in the CESM2 for annual updates of the global ITS. In the context of increasing global heat stress, this research topic has received much attention recently, and this study highlights the importance of sustainable socio-economic pathways and efficient irrigation technologies to provide more solid evidence on those issues. The manuscript is clearly written, the results are well presented through several high-quality tables and graphs, overall I would recommend the manuscript for publication in Nature Communications but have a few comments that might be considered in a revised version of the manuscript:

Major comments:

1-Current study was conducted based on a combination of SSP1, SSP3 with RCP2.6, 7.0 scenarios, and the fact that the findings are very sensitive to the choice of SSP-RCP scenarios, a discussion of the uncertainty in the results due to the choice of different pathways seems to be necessary.

2-I agree with the key hypothesis that “richer and drier countries will invest more in high-efficient irrigation”. For the East Asia region, with the irrigated area remaining essentially unchanged, IWW has been reduced by about 30-40% through improvements in irrigation water use efficiency as irrigation technology has evolved (estimated from Figure 2). In East Asia, China has the most potential and motivation to improve irrigation efficiency. In the case of China, for example, the current irrigation water use efficiency is about 0.56, and the results in the manuscript suggest that the water use efficiency would need to be raised to nearly 1 in the future, which confuses me. Consideration of irrigation efficiency threshold constraints in the ITS dataset seems necessary, if possible.

3-In this study, it is assumed that the evolution of drip and sprinkler irrigation to replace flood irrigation is at a constant rate of 1%, and as you mentioned, paddy irrigation is included in flood irrigation, and planting rice with drip and sprinkler irrigation techniques seems technically challenging currently. Especially in East Asia, where rice cultivation is dominant, will this underestimate the irrigation impacts on extreme dry and humid heat in future scenarios?

4-CESM2 has no limits on irrigation withdrawals for the irrigation module, and in fact irrigation withdrawals are limited by the regional river flows, and for the future new hotspots mentioned in the manuscript, whether the unconstrained application of irrigation withdrawals would overestimate the irrigation impacts on the extreme heat in the region represented by the Middle East and Africa.

5-Lines 219-222 illustrate that irrigation practices in traditionally highly irrigated areas have minor impacts on extreme dry heat under the future scenarios, this point deserves more in-depth comparative analysis and discussion. The comparisons in Figures 4c and f seem to compare extreme dry heat differences between different SSP-RCP pathways rather than the impact of irrigation.

6-As you mentioned in line 315, some areas that are currently considered unsustainable for irrigation practices continue to experience an increase in irrigation water withdrawal due to irrigation expansion under future scenarios, and their viability deserves further validation and discussion.

7-This study is a continuation of the authors' recent achievement (DOI:10.1038/s41467-025-56356-1), which has in-depth insights in this field, and I believe that if they could consider how to guide the actual irrigation patterns towards the most beneficial direction for human based on the previous quantitative analysis and mechanism exploration studies of irrigation impacts on heat stress in the discussion section, it will provide a solid theoretical basis for the climate change adaptation strategies in agriculture.

Minor comments:

1-The abbreviation of Figure 4 on comparing the differences between future and historical scenarios fails to convey a clear enough meaning.

2-Provide formulas for key evaluation indicators, such as wet bulb temperature, in the methodology section for better independent understanding.

3-The figures and tables in the Appendix would be better as Support Information for the manuscript.

4-Driving CESM2 simulations using SSP-RCP scenarios and innovative ITS datasets is the core of this study, and the necessary descriptions of the CESM2 modeling related setup in the Methods section (or support information) will make the paper more perfect.

Liangfeng Zou (PhD student) and Yuanyuan Zha (Associate Professor) at Wuhan University

(Remarks on code availability)
the code is properly provided.

Reviewer #3

(Remarks to the Author)

The study “Compounding future escalation of emissions- and irrigation-induced increases in humid-heat stress” discusses different irrigation scenarios and their impact on various heat events. It is a valuable contribution to understanding the feedback between irrigation methods and local heat stress. I thoroughly enjoyed reading the paper. I have a few comments that may help expand the discussion, which are listed below:

1) What if the assumed ITS transition rate of 1% per year is not accurate? Can you briefly describe what would happen if the ITS rate were different, and what the plausible scenarios might be?

2) Doubling the number of cropland columns from 64 to 128 increases model complexity and computational cost. If the data used to allocate irrigation methods across columns is uncertain or based on assumptions, how do you ensure that this added complexity results in more reliable outcomes? Can you also discuss the limitations of the model used?

3) Could you elaborate on the assumptions underlying the hydro-economic framework? For instance:

i) Classifying countries by national-level variables misses within-country variability in both irrigation practices and driving factors

ii) Since the driver selection is based on Spearman's correlation, how do you address the fact that correlation does not imply causation?

- iii) Why is precipitation used as the sole indicator of hydro-climatic conditions, and how do you account for other key factors like groundwater availability, river discharge, infrastructure, or water rights?
- iv) The framework is static and scenario-based: it projects future ITS based on fixed drivers but does not simulate feedbacks (e.g., how changes in ITS might influence GDP, water stress, or governance over time).

(Remarks on code availability)

Version 1:

Reviewer comments:

Reviewer #1

(Remarks to the Author)

Review of "Compounding future escalation of emissions- and irrigation-induced increases in humid-heat stress" submitted to Nature Communications for consideration for publication.

The manuscript has been revised and many improvements have been made. This includes a restructuring and better description of the results. Moreover, the discussion section has been revised and restructured. These changes have increased the readability of the manuscript. Apart from the minor comments below, I think the manuscript is acceptable for publication.

Below I have provided specific comments to the text, figures and tables.

Specific comments

Lines 41-45: Please revise this sentence. I think the previous version said correctly "increases", which now was changed to "increase" (line 42). Moreover, it is unclear to what "them" refers to in line 43.

Lines 93-96: I still don't understand why the authors consider the SSP5 scenario here. The main document does not include any reference to this scenario, so why mention it here? Again, I highly suggest to remove the SSP5 scenario altogether from the manuscript (also from the Methods section).

Line 180: Please replace "is increased" by "has increased".

Line 182: I'm not sure if "intensify" is correctly used here. Maybe better say "increase" or "amplify".

Lines 241-244: These results consider the traditional hotspot areas? Please indicate in the text.

Lines 249-250: I'm not so convinced about the difference between the scenarios with and without irrigation. In Figure 7 these differences are not that big for most of the data shown. Only for dry-heat for the traditional irrigation hotspots this difference seems rather obvious, but for the other scenarios not that much. Please change this sentence accordingly.

Lines 260-264: Please explicitly indicate which assumption belongs to which SSP scenario.

Lines 327-369: These three paragraphs include many limitations to the presented methods. The authors propose some changes to the existing method. However, it would be interesting to read how these proposed methods would affect the results. So, instead of including a longer list of improvements, it would be interesting to select the most relevant and describe how these improvements might affect the results.

Lines 371-376: I suggest to end the Discussion section with a paragraph with some concluding remarks, instead of the limitations of the study.

(Remarks on code availability)

The authors included comments to the code so it can be better understood by the interested readers.

Reviewer #2

(Remarks to the Author)

We think the revision is detailed and effective. So we recommend acceptance of the manuscript.

Yuanyuan Zha and Liangfeng Zou at Wuhan University

(Remarks on code availability)

We checked the code last time.

Reviewer #3

(Remarks to the Author)

The authors have provided substantial additional discussion in the manuscript, effectively addressing several of the queries I raised. The revised version presents a clearer and more comprehensive picture of the work undertaken and the aspects to be addressed in the future. I am satisfied with the manuscript in its current form.

(Remarks on code availability)

Compounding future escalation of emissions- and irrigation-induced increases in humid-heat stress

July 5, 2025

Yi Yao^{*1,2}, Yusuke Satoh³, Nicole van Maanen⁴, Sabin Taranu¹, Jessica Keune⁵, Steven J. De Hertog^{6,1}, Sepepe Lampe¹, David M. Lawrence⁷, William J. Sacks⁷, Yoshihide Wada⁸, Agnès Ducharne⁹, Benjamin I. Cook^{10,11}, Sonia I. Seneviratne², Laibao Liu^{12,13}, Jonathan R. Buzan¹⁴, Jonas Jägermeyr^{15,16,17}, Wim Thiery¹
yi.yao@vub.be

¹ Department of Water and Climate, Vrije Universiteit Brussel, Brussels, Belgium

² Institute for Atmospheric and Climate Science, Eidgenössische Technische Hochschule Zürich, Zurich, Switzerland

³ Japan Agency for Marine-Earth Science and Technology, Yokohama, Japan

⁴ Department of Water and Climate Risk, Institute for Environmental Studies (IVM), Vrije Universiteit Amsterdam

⁵ Forecast and Services Department, European Center for Medium-Range Weather Forecasts, Bonn, Germany

⁶ Q-ForestLab, Department of Environment, Universiteit Gent, Ghent, Belgium

⁷ NSF National Center for Atmospheric Research, Boulder, Colorado, USA

⁸ Biological and Environmental Science and Engineering Division, King Abdullah University of Science and Technology (KAUST), Saudi Arabia

⁹ Laboratory 7619 METIS, Sorbonne Université, CNRS, EPHE, IPSL, Paris, France

¹⁰ NASA Goddard Institute for Space Studies, New York, New York

¹¹ Lamont-Doherty Earth Observatory, Palisades, New York

¹² Department of Geography, The University of Hong Kong (HKU), Hong Kong

¹³ Institute for Climate and Carbon Neutrality, The University of Hong Kong (HKU), Hong Kong

¹⁴ Department of Chemistry and Bioscience, Aalborg University, Aalborg, Denmark

¹⁵ Columbia University, Climate School, New York, USA

¹⁶ NASA Goddard Institute for Space Studies, New York, NY, USA

¹⁷ Potsdam Institute for Climate Impact Research (PIK), Member of the Leibniz Association, Potsdam, Germany

Contents

1	Reviewer 1	4
1.1	Major comments	4
	Reviewer 1 Comment 1	4
	Reviewer 1 Comment 2	4

Reviewer 1 Comment 3	5
Reviewer 1 Comment 4	6
Reviewer 1 Comment 5	8
1.2 Minor comments	9
Reviewer 1 Comment 6	9
Reviewer 1 Comment 7	9
Reviewer 1 Comment 8	9
Reviewer 1 Comment 9	9
Reviewer 1 Comment 10	9
Reviewer 1 Comment 11	9
Reviewer 1 Comment 12	11
Reviewer 1 Comment 13	13
Reviewer 1 Comment 14	13
Reviewer 1 Comment 15	13
Reviewer 1 Comment 16	13
Reviewer 1 Comment 17	14
1.3 Remarks on code availability	15
Reviewer 1 Comment 18	15
2 Reviewer 2	16
2.1 Major comments	16
Reviewer 2 Comment 1	16
Reviewer 2 Comment 2	16
Reviewer 2 Comment 3	17
Reviewer 2 Comment 4	19
Reviewer 2 Comment 5	20
Reviewer 2 Comment 6	20
Reviewer 2 Comment 7	20
Reviewer 2 Comment 8	21
2.2 Minor comments	22
Reviewer 2 Comment 9	22
Reviewer 2 Comment 10	22
Reviewer 2 Comment 11	22
Reviewer 2 Comment 12	22
3 Reviewer 3	23
3.1 Major comments	23
Reviewer 3 Comment 1	23
Reviewer 3 Comment 2	23
Reviewer 3 Comment 3	23
Reviewer 3 Comment 4	25

Abstract

This response letter contains numbered figures and tables and references to them. To prevent confusion, the figures and tables embedded within this response letter are called FIGs and TABs, while the figures and tables embedded within the manuscript are still called Figures and Tables. The following convention is applied to denote modification in the original manuscript: new text.

1 Reviewer 1

1.1 Major comments

Reviewer 1 Comment 1

The manuscript describes a global study on the impact of climate change on irrigation development under 2 SSPs. The authors developed a new irrigation module for CESM2, which now considers more irrigation techniques. The model is then applied under a historical scenario and for different SSPs in the 21st century. The results show that irrigation demand will decrease under SSP1 and increase under SSP3. Based on the results of the simulations, the authors also determined the impacts on dry-heat and moist-heat stress. These results show moderate changes for dry-heat stress, but substantial increases for moist-heat stress under the different SSPs. The study seems a valuable addition to the literature, with novel results related to the projection of irrigation for the 21st century, including the impacts on dry-heat and moist-heat stress.

Response We thank Reviewer 1 for these encouraging comments and the valuable feedback, which has helped us to improve our manuscript. Below, we address every comment carefully and explain the corresponding changes in the manuscript.

Reviewer 1 Comment 2

There are two issues regarding the considered SSPs that would need some more explanation in the manuscript. The authors mainly focus on two SSPs, i.e. SSP1 and SSP3, while SSP5 is also considered, but not fully included in the results. This is somewhat confusing. Please provide the reasoning why SSP5 is not fully considered. In either case, it would make sense to either exclude SSP5 from the manuscript all together or better integrate the SSP5 results in the manuscript.

Response Thanks for pointing this out. We are sorry for the confusion it creates. In this study, the irrigation techniques share data was generated under all three scenarios (SSP1-2.6, SSP3-7.0 and SSP5-8.5), because we have access to the SSP and ISIMIP datasets under these three scenarios. Regarding the simulations, as we were limited by computational resources, we selected the two most extreme scenarios (SSP1-2.6: both low challenges; and SSP3-7.0: both high challenges) (1) out of three to proceed with. To avoid the confusion, we fully exclude SSP5 from the main text and move it to the Supplementary Materials (see Supplementary Figures 2.2). Some texts are also added to let readers know that the dataset under SSP5-8.5 has also been created.

L93-96

Two SSP-RCPs, SSP1-2.6 (the green road, with low challenges for both mitigation and adaptation) and SSP3-7.0 (the rocky road, with both high challenges), are selected (the dataset under SSP5-8.5 is also generated, see Supplementary Information, but is not used further in the simulations).

L624-628

Given the data availability of the hydro-climatic data, we choose SSP1 (the green road: low in both adaptation and mitigation challenges), SSP3 (the rocky road: high in both challenges), and SSP5 (the highway: high in challenges for mitigation and low in challenges for adaptation) to generate the data (SSP5 not used in simulations).

Reviewer 1 Comment 3

In addition, I was not expecting that SSP3 would result in a such a substantial increase in irrigation. Following O'Neill et al. (2017), it seems that SSP5 is a more likely pathway that would result in such an increase in irrigation, given the assumption that agriculture will be "Highly managed, resource-intensive; rapid increase in productivity" under SSP5. Agriculture under SSP3 is suggested to be characterized by "Low technology development, restricted trade", which does not suggest such a high focus on irrigation, especially drip irrigation. The description of how the different SSPs affect irrigation development fall short in this respect (lines 573-586). Hence, the authors should better explain how the different SSPs are implemented and discuss in the main document how the implementation of the SSPs compare to the literature, e.g. O'Neill et al. (2017). O'Neill, B.C., Kriegler, E., Ebi, K.L., Kemp-Benedict, E., Riahi, K., Rothman, D.S., van Ruijven, B.J., van Vuuren, D.P., Birkmann, J., Kok, K., Levy, M., Solecki, W., 2017. The roads ahead: Narratives for shared socioeconomic pathways describing world futures in the 21st century. *Global Environmental Change* 42, 169–180. <https://doi.org/10.1016/j.gloenvcha.2015.01.004>

Response We thank Reviewer 1 for this constructive suggestion of expanding the description of the impacts of different SSPs on irrigation development. We agree that in the previous version, the description was not comprehensive. In this study, the upgrade rate of irrigation equipment is totally dependent on the socio-economic capacities and hydro-climatic conditions.

The variables representing socio-economic capacities are obtained from the SSP dataset, and the outcomes align with the narratives of SSP. In TAB 1 we can find the assumptions regarding land use and agriculture.

TAB 1: Assumptions regarding land-use and agriculture of SSP narratives (1; 2)

SSPs	Land use	Agriculture	Irrigation
SSP1	Strong regulations to avoid environmental tradeoffs	Improvements in ag productivity; rapid diffusion of best practices	Large improvements in irrigation water use efficiency where possible
SSP3	Hardly any regulation; continued deforestation due to competition over land and rapid expansion of agriculture	Low technology development, restricted trade	Only modest improvements in irrigation water use efficiency
SSP5	Medium regulations lead to slow decline in the rate of deforestation	Highly managed, resource-intensive; rapid increase in productivity	Large improvements in irrigation water use efficiency

SSP3 represents a rocky road with regional rivalry, under which the gap between developed and developing countries grows bigger. The population is expected to grow at a high rate in developing countries, increasing the global food demand. At the same time, the priority for addressing environmental issues is low, so the irrigation is assumed to increase with a low rate of efficiency increase. SSP5 represents a pathway with rapid economic and social development. Environmental issues are expected to be solved via technological solutions in the second half of the century. It is confusing that irrigation extent experiences a rapid increase under SSP3, even though it is designed to be low-managed. The reason for the confusion is that in our study, we do not see flood irrigation as a highly-managed activity. Instead, flood irrigation could be achieved with little investment in equipment, and it should be seen as a very inefficient way of water-use.

We also added some text in the Discussion Section: L260-273

The assumptions regarding land use and agriculture under SSP1 and SSP3 are provided as "strong regulations to avoid environmental tradeoffs; improvements in agriculture productivity; rapid diffusion of best practices", and "Hardly any regulation; continued deforestation due to competition over land and rapid expansion of agriculture; Low technology development, restricted trade". The LUH2 dataset agrees with those narratives, with the key feature that AEI keeps almost constant under SSP1-2.6 and expands rapidly under SSP3-7.0 (Figure 1, 2). Some assumptions regarding the irrigation water use efficiency have also been proposed, with "large improvements in irrigation water use efficiency where possible" for SSP1 and "only modest improvements in irrigation water use efficiency" for SSP3. The generated ITS data align well with those assumptions, with a rapid decrease in traditional inefficient flood irrigation under SSP1-2.6 and a slower decrease under SSP3-7.0 (Figure 1, 3). Although we apply a simplified adjustment factor to account for annual ITS changes based on socio-economic and hydro-climatic variables (see Methods), the resulting estimates are consistent with prevailing assumptions, thereby offering a valuable enhancement to existing datasets.

Reviewer 1 Comment 4

Considering the results, the authors show the results for dry-heat and moist-heat stress for scenarios with and without irrigation. I am not fully convinced of the added value of showing the results without irrigation, at least in the main document. The first part of the Results section solely focuses on how irrigation is projected to change under the different SSPs. In the second part the authors also consider scenarios without irrigation. Although these scenarios can show how heat stress is affected by climate change only, they are not realistic scenarios, but more “what if” scenarios. I suggest to move the scenarios without irrigation to the Supporting Material and mostly focus on the scenarios with irrigation, also when presenting the results in the text. This would significantly improve the readability of the text, where especially lines 176-222 are sometimes difficult to follow because of the number of different comparisons that are made by the authors, i.e. historical vs SSP and irrigation vs no irrigation between the SSPs and between historical and the SSPs. In that sense, for Figures 4 and 5, I would keep panels a, c and d in the main document, while b, e and f can be moved to the Supporting Material.

Response

We thank Reviewer 1 for this suggestion. We changed the structure of the Results section to discuss the impacts of climate change and irrigation separately, which can help avoid confusion. We also moved all results without irrigation to the Supplementary Materials. In the old version of the manuscript, we structured the Results section as follows.

- 1. Irrigation water withdrawal highly depend on projected irrigation expansion and efficiency**
- 2. No more reversal of dry-heat extremes in irrigation hot spots**
- 3. Intensified local moist-heat stress**

After introducing the changes in the area equipped for irrigation and irrigation techniques share, we described the changes in dry- and moist-heat extremes separately, in which we talked about the impacts of different forcings, and focused on both spatial distribution and time series, which could be confusing.

To address this issue, in the new version of the manuscript, we structure the Results section as follows.

- 1. Irrigation water withdrawal highly depend on projected irrigation expansion and efficiency**
- 2. Divergent impacts of irrigation on dry- and moist-heat extremes**
- 3. Rapid increase in exposure to heat extremes caused by climate change**
- 4. Difference between dry- and moist-heat extremes hours increase enlarged by irrigation**

We also changed Figure 4, Figure 5, and Figure 6, which are now respectively the figures for impacts of irrigation, impacts of all forcings, and time series.

We now first introduce the impacts of irrigation on moist-heat extremes (subsection 2), and then describe the impacts of all forcings (subsection 3), and finally, we compared the impacts of irrigation and other forcings (subsection 4). We hope the new version can avoid the confusion.

Reviewer 1 Comment 5

The concluding remarks focus mostly on the suggestion to move to an SSP1 world (lines 321-330 and lines 341-353). From the perspective of sustainability, the goal to decrease irrigation water use and the impacts on heat stress, I fully agree with this. But is this also a realistic goal? Since the authors considered only two SSPs, which are in most aspects also their opposites, it seems logic that the authors come to the conclusion that SSP1 would be better aim. However, it is likely that other SSPs may be more realistic given the ongoing socioeconomic and climatic developments. More discussion is needed that would need the involvement of the other SSPs, given their particularities and possible impacts on irrigation and heat stress.

Response We fully agree with Review 1 on this point, so we changed some texts to stress that the results of this study indicate the importance of increasing the irrigation efficiency and avoiding unsustainable irrigation expansion, rather than focusing on SSP1, which is a highly idealised and outdated scenario.

We have added some text in the discussion part.

L251-258

In this study, we, for the first time, design ITS scenarios aligned with the SSP-RCPs, improve an Earth system model to incorporate this information, and perform fully-coupled climate simulations under varying climate and irrigation transition pathways. We then analyse the simulation outputs for irrigation water demand and irrigation-induced impacts on dry and moist-heat stress. The results exhibit a strong sensitivity to the choice of SSP-RCP scenarios, highlighting their important implications for future irrigation planning and development. While these scenarios rely on several idealised assumptions, the underlying narratives still offer valuable insights for guiding future irrigation strategies.

L371-376

Generally, the results exhibit a strong dependence on the selected scenarios, with this study focusing exclusively on SSP1 and SSP3. These scenarios represent two contrasting extremes in terms of climate mitigation and adaptation risks. However, they may no longer reflect the most current socioeconomic and climate projections, as newer scenarios have since been developed to address contemporary research questions. Future irrigation-related studies should therefore incorporate updated scenario frameworks to yield more relevant insights for policy and planning.

1.2 Minor comments

Reviewer 1 Comment 6

Lines 31-32: With "...its climatic impacts in future with possible changes in both extent and efficiency considered" the authors mean "... its future climatic impacts considering possible changes in both extent and efficiency".

Response We thank Review 1 for the suggestion to rephrase this sentence. It has been changed in the manuscript.

L30-32

Irrigation has been investigated as an important historical climate forcing, but there is no study exploring its future climatic impacts considering possible changes in both extent and efficiency.

Reviewer 1 Comment 7

Line 43: The authors mean ≥ 100 hours yr-1? Please revise.

Response Thanks for finding the missing unit. We have added the unit.

Reviewer 1 Comment 8

Lines 93-94: Why is it relevant to mention that SSP5-8.5 is also simulated but not considered? Please clarify in the text.

Response We now move all materials regarding SSP5-8.5 to the Supplementary Materials to avoid confusion.

Reviewer 1 Comment 9

Lines 96-98: The text between the parentheses depends on the socioeconomic scenario, right? Please clarify this in the text.

Response Thanks for pointing it out. We have added some text there.

L98-100

A basic assumption is that countries with higher socio-economic capacities and drier climate conditions have more motivation to upgrade their irrigation system, which vary among scenarios.

Reviewer 1 Comment 10

Figure 1: Please revise the figure caption, where it says two times "equipped", while one time seems sufficient.

Response Thanks for this suggestion. We have deleted one "equipped" there.

Reviewer 1 Comment 11

Figure 2d-2g: It is difficult to understand where the historical IWW results are shown. I suggest to make the grey bar as wide as the historical period (1985-2014), to make this stand out more. Moreover, I don't see the point of the vertical green and red lines for the two SSP scenarios. I suggest to remove these lines and only include the trend line.

Response

Thanks for this advice! We have replotted the figure (see FIG 1.)

FIG 1: Historical irrigation water withdrawal and projected changes under SSP1-2.6 and SSP3-7.0 scenarios. **a** Annual mean irrigation water withdrawal (IWW) in the historical period (1985-2014). **b,c** Projected changes in annual mean IWW in future period (2045-2074) under SSP1-2.6 (**b**) and SSP3-7.0 (**c**), compared to the historical period. The values shown here are the mean values of all three ensemble members. **d-g** Global (**d**) and sum of regional annual IWW of three groups of regions (**e-g**) during the historical (1985-2014) and future (2015-2074) periods under SSP1-2.6 and SSP3-7.0. Historical IWW is visualised as a single bar in the year 2000, as in the historical simulations, AEI and ITS are static. The bar indicates the mean value and the range indicates the maximum and minimum values of the 30-year period. In future periods darker bars indicate the range simulated by three ensemble members and lines indicate the median values. All values in the future period have been smoothed using Savitzky-Golay filtering (order = 2, window = 7). Regions of each group are listed in Table S1, and their locations are visualised in Figure a. IWW of individual regions are visualised in Figure S6.

Reviewer 1 Comment 12

Lines 131-151: The results of Figure 2 are presented here, going progressively from panel a to g. I suggest to integrate better the results of panels a-c and panels d-g. For instance, by presenting first the results of SSP1 and discussing simultaneously the results of panels d-g for this scenario, etc.

Response Thanks for this suggestion! We have restructured this part of results. Now we discuss the results of SSP1 and SSP3 separately. We hope it can avoid the confusion.

L112-158

The Land Use Harmonization version 2 (LUH2) dataset shows that under SSP1-2.6, the global AEI remains nearly constant at around $2.7\text{--}2.8 \times 10^6 \text{ km}^2$, with a rapid increase in the projected use of efficient irrigation techniques according to our dataset (drip and sprinkler accounting for $\sim 2/3$ in 2100, up from $\sim 1/9$ in 2015) (Figure 1a). Under SSP3-7.0, projected irrigated land³¹ increases steadily, reaching more than $4 \times 10^6 \text{ km}^2$ by the end of the century, with a smaller fraction of drip and sprinkler techniques (less than $1/2$ in 2100) (Figure 1b).

Based on the spatial distribution of AEI and ITS (Figure 2, 3, and S1-2), we select 13 IPCC reference regions³² with high irrigation activity or with varying changes in AEI under two scenarios (Figure 4a), and pool them into three groups (Table S1). Group 1 includes regions (solid line in Figure 2a) with higher socio-economic capacities, like Central North America and East Asia, enabling AEI to remain nearly constant and efficient irrigation to grow rapidly under both scenarios (slightly faster under SSP1-2.6, see Figure 1c,d and S3). Group 2 consists of the regions (dashed line in Figure 2a) with large AEI but lower socio-economic capacities in present-day, such as South Asia and West Central Asia. In Group 2, AEI remains relatively constant under SSP1-2.6 but expands rapidly under SSP3-7.0 (Figure 1e,f and S4). The regions in Group 3 (dotted line in Figure 2a) have almost no AEI historically and under SSP1-2.6, but their AEI experiences substantial expansion under SSP3-7.0, making them irrigation hot spot regions by the end of the century. These future hotspot regions for example include West and South Africa (Figure 1g,h and S5).

IWW is highly related to spatial and temporal changes in both AEI and ITS (Figure 4). From 1985 to 2014, North India was the most intensely irrigated area with simulated IWW surpassing 500 mm yr^{-1} in some grid cells (over the entire $0.9^\circ \times 1.25^\circ$ grid cell area). Other heavily irrigated areas included East China and Central USA, where simulated IWW also exceeds 300 mm yr^{-1} in several grid cells (Figure 4a). Globally, IWW ranges between $1700 - 2000 \text{ km}^3 \text{ yr}^{-1}$ during the historical period (in historical simulations, AEI and ITS is static at the level of the year 2010), and IWW projections start from around $2100 \text{ km}^3 \text{ yr}^{-1}$ in the year 2015 (Figure 4d).

IWW is projected to decrease in many grid cells over Central North America and East Asia under SSP1-2.6 (2045-2074) compared to the historical period (1985-2014), where the reduction exceeds 50 or even 100 mm yr^{-1} locally (Figure 4b). Global IWW decreases slightly to $\sim 1700 \text{ km}^3 \text{ yr}^{-1}$ by 2074 (Figure 4d), with the regional values falling from ~ 400 to $\sim 300 \text{ km}^3 \text{ yr}^{-1}$ in Group 1 (Figure 4e and S6a-c), from ~ 1300 to $\sim 1100 \text{ km}^3 \text{ yr}^{-1}$ in Group 2 (Figure 4f and S6d-h), and remaining constant at around $50 \text{ km}^3 \text{ yr}^{-1}$ in Group 3 (Figure 4g and S6i-m).

Under SSP3-7.0 (2045-2074), the reduction in IWW persists in Central North America and East Asia, but new irrigation hot spots appear across Africa, and IWW increases in some traditional irrigation hot spots like in South Asia (Figure 4c). After 2015, IWW increases continuously to $\sim 2400 \text{ km}^3 \text{ yr}^{-1}$ under SSP3-7.0 (Figure 4d). IWW shows different trends across the three region groups due

to the varying socioeconomic developments in each group. In Group 1, IWW decreases to less than $300 \text{ km}^3 \text{ yr}^{-1}$ (Figure 4e and S6a-c), owing to a steady AEI and improvements in ITS (Figure 1c,d and S3). In Group 2, IWW increases to $\sim 1400 \text{ km}^3 \text{ yr}^{-1}$ (Figure 4f and S6d-h), mainly attributed to the expanding AEI (SSP3-7.0) (Figure 1e,f and S4). In Group 3, a sharp increase from 50 to $400 \text{ km}^3 \text{ yr}^{-1}$ in IWW is projected (Figure 4g and S6i-m) due to a rapid AEI expansion (Figure 1g,h and S5).

Reviewer 1 Comment 13

Lines 282-288: I would say that farmers' individual decision-making and government regulation are part of the socioeconomic conditions. But I can be wrong.

Response Yes, we agree. Farmers' individual decision-making and government regulation can be largely affected by socioeconomic conditions. We therefore have modified the sentence.

L342-345

In reality, farmers' individual decision-making and government policies are **not fully determined by socio-economic capacities**. For example, farmers' decisions to improve irrigation efficiency are motivated by multiple factors, especially the cost and benefits.

Reviewer 1 Comment 14

Lines 270-305: Limitations are usually discussed at the end of the Discussion section.

Response We thank Reviewer 1 for this advice. The Discussion Section has been restructured, and now the limitations are the last part of it.

Reviewer 1 Comment 15

Lines 583-586: Please explain here why the authors focused on these two extreme SSPs, i.e. SSP1 and SSP3 (not considering SSP5). Moreover, in the main text the authors focus on SSP1 and SSP3. It is a bit confusing that results are shown for SSP5 also, but not really considered in the main document. I suggest to remove this from the manuscript or incorporate the results in the main text.

Response We apologise for this confusion. We have moved the results under SSP5-8.5 from main texts to Supplementary Information.

Reviewer 1 Comment 16

Lines 620-632: I suggest to include a flow chart to make this easier to understand.

Response Thank for this comment. The flowchart has been added in Supplementary Figures (see FIG 2), and a paragraph representing an example is given.

L679-693

FIG 2: Illustrative flowchart for irrigation techniques share change.

Given the complexity of the process, a representative example is presented for illustrative purposes (a flowchart is also given as shown in Figure S8). In the year 2075, a grid cell is assumed to contain 70% flood irrigation, 20% sprinkler irrigation, and 10% drip irrigation. The crop type distribution is assumed to be 35% flood crops, 50% sprinkler crops, and 15% drip crops (see Table S4). The procedure begins by calculating the normalised 20-year average precipitation for the period 2056–2075, which is assumed to be 0.45. Subsequently, the first principal component of the socio-economic indicators—GDP, GOV, URB, and GII—is computed using the PCA model previously developed from historical SSP datasets, yielding a normalised value of 1.25. The annual updating rate is determined by subtracting the hydro-climatic adjusting factor (0.1) from 1 and adding the socio-economic adjusting factor (0.8) (Table S4), resulting in a rate of 1.7% per year over the subsequent five-year period (until 2080) (Table S5). Accordingly, by 2080, the flood irrigation fraction is reduced by 8.5% ($1.7\% \times 5$ years), with the reduction redistributed to sprinkler and drip irrigation in proportions of 6.54% and 1.96%, respectively. Thus, by 2080, the shares of flood, sprinkler, and drip irrigation are adjusted to 61.5%, 26.54%, and 11.96%, respectively.

Reviewer 1 Comment 17

Line 671: Please give the definition of NOI.

Response We apologise for the confusion. We have rewritten this sentence.

L725-727

We first conduct the two historical simulations with (Hist_IRR) and without irrigation (Hist_NOI) covering the period 1980-2014, with first five years as the spin-up period, using the land use and land management of the year 2010, and including the existing ITS dataset (Figure 8).

1.3 Remarks on code availability

Reviewer 1 Comment 18

A README file is provided, but only gives limited information on what the different scripts accomplish.

In general, most scripts do not include comments on the functionality of the script. It is therefore difficult to understand what the code is meant for. Moreover, the script are not particularly DRY (Don't Repeat Yourself), there is a lot of repetition, which makes the code difficult to understand. Also the code is written in three different programming languages (Matlab, Python and shell script), which is of course not ideal.

Response Thanks for pointing it out! We have rewritten the codes in Python, improved the codes to avoid repetitions, and also uploaded the codes developed for CESM2 and for plotting. Please check the github repository https://github.com/YiYao1995/Yao_et_al_2025_Nature_Communications_Compounding_Future_Escalation.git.

2 Reviewer 2

2.1 Major comments

Reviewer 2 Comment 1

This is an interesting article that investigates the irrigation impacts on extreme dry and humid heat under future scenarios of changes in irrigation extent and irrigation efficiency. The authors set up different SSP-RCP pathway scenarios to develop dynamic global irrigation techniques share (ITS) datasets and embed them in the CESM2 Model, and conduct sensitive simulation experiments with CESM2 to assess irrigation water withdraw and quantify irrigation impacts on extreme dry and humid heat under dynamic conditions of changing irrigation extent and efficiency. The innovation of this study is the development of the dataset that considers irrigation technology change using a combination of SSP and ISIMIP3 scenarios embedded in the CESM2 for annual updates of the global ITS. In the context of increasing global heat stress, this research topic has received much attention recently, and this study highlights the importance of sustainable socio-economic pathways and efficient irrigation technologies to provide more solid evidence on those issues. The manuscript is clearly written, the results are well presented through several high-quality tables and graphs, overall I would recommend the manuscript for publication in Nature Communications but have a few comments that might be considered in a revised version of the manuscript:

Response

We sincerely thank Reviewer 2 for the encouraging feedback and valuable suggestions, which have contributed to improving our manuscript. Below, we provide detailed responses to each comment.

Reviewer 2 Comment 2

Current study was conducted based on a combination of SSP1, SSP3 with RCP2.6, 7.0 scenarios, and the fact that the findings are very sensitive to the choice of SSP-RCP scenarios, a discussion of the uncertainty in the results due to the choice of different pathways seems to be necessary.

Response Indeed, the results are strongly sensitive to the choice of SSP-RCP scenarios, so we added some texts in the main text.

L251-258

In this study, we, for the first time, design ITS scenarios aligned with the SSP-RCPs, improve an Earth system model to incorporate this information, and perform fully-coupled climate simulations under varying climate and irrigation transition pathways. We then analyse the simulation outputs for irrigation water demand and irrigation-induced impacts on dry and moist-heat stress. The results exhibit a strong sensitivity to the choice of SSP-RCP scenarios, highlighting their important implications for future irrigation planning and development. While these scenarios rely on several idealised assumptions, the underlying narratives

still offer valuable insights for guiding future irrigation strategies.

L371-376

Generally, the results exhibit a strong dependence on the selected scenarios, with this study focusing exclusively on SSP1 and SSP3. These scenarios represent two contrasting extremes in terms of climate mitigation and adaptation risks. However, they may no longer reflect the most current socioeconomic and climate projections, as newer scenarios have since been developed to address contemporary research questions. Future irrigation-related studies should therefore incorporate updated scenario frameworks to yield more relevant insights for policy and planning.

Reviewer 2 Comment 3

I agree with the key hypothesis that “richer and drier countries will invest more in high-efficient irrigation”. For the East Asia region, with the irrigated area remaining essentially unchanged, IWW has been reduced by about 30-40% through improvements in irrigation water use efficiency as irrigation technology has evolved (estimated from Figure 2). In East Asia, China has the most potential and motivation to improve irrigation efficiency. In the case of China, for example, the current irrigation water use efficiency is about 0.56, and the results in the manuscript suggest that the water use efficiency would need to be raised to nearly 1 in the future, which confuses me. Consideration of irrigation efficiency threshold constraints in the ITS dataset seems necessary, if possible.

Response We agree that this is a very confusing point. In this study, we do not directly project the change in irrigation water use efficiency, but rather the relative fraction of three different irrigation techniques. In CESM2, the hydrological processes after the application of irrigation are simulated by its hydrology module, which means that the irrigation efficiency is dependent on many factors, like background climate, soil type, crop type, etc. Even for the same irrigation technique, the irrigation efficiency can be different. In this study, following Jagermeyr et al. (2015) (3), we divided all crop types into three groups based on the suitability of irrigation techniques (TAB 2), which kind of already defines the upper threshold of high-efficiency irrigation (sprinkler and drip).

At the same time, the increase in irrigation efficiency can also be triggered and boosted by other factors rather than socio-economic and hydro-climatic conditions. The classification framework used in this study is very idealised, so we also added some texts in the discussion section.

L341-353

Regarding the assumptions applied for transitions in ITS, only socioeconomic and hydro-climatic conditions are considered as drivers. In reality, farmers' individual decision-making and government policies are not fully determined by socio-economic capacities. For example, farmers' decisions to improve irrigation efficiency are motivated by multiple factors, especially the cost and benefits. Another issue is that the socio-economic capacity variables are assumed to be the

TAB 2: Suitable irrigation techniques for different crop types in CESM2 input data

Crop type	drip	sprinkler	flood
Spring wheat	No	Yes	Yes
Winter wheat	No	Yes	Yes
Barley	No	Yes	Yes
Winter barley	No	Yes	Yes
Rye	No	Yes	Yes
Winter rye	No	Yes	Yes
Rice	No	No	Yes
Temperate maize	No	Yes	Yes
Tropical maize	No	Yes	Yes
Millet	No	Yes	Yes
Sorghum	No	Yes	Yes
Pulses	Yes	Yes	Yes
Cassava	No	No	No
Sunflower	Yes	Yes	Yes
Temperate soybean	Yes	Yes	Yes
Tropical soybean	Yes	Yes	Yes
Groundnuts	No	Yes	Yes
Rapeseed	No	Yes	Yes
Sugarcane	No	Yes	Yes
Citrus	Yes	Yes	Yes
Cocoa	Yes	Yes	Yes
Coffee	Yes	Yes	Yes
Cotton	Yes	Yes	Yes
Date palm	Yes	Yes	Yes
Grapes	Yes	Yes	Yes
Oil palm	Yes	Yes	Yes
Potatoes	Yes	Yes	Yes
Fodder grass	No	Yes	Yes
Miscanthus	No	Yes	Yes

same within each country, ignoring the internal variability within bigger countries. Moreover, a 1% default rate of flood irrigation reduction is assumed for all countries based on historical time series of two countries (see Method), which may not be realistic for other countries, and changing this default rate may lead to different results. Furthermore, precipitation is the only variable used to indicate hydro-climatic complexity, but some other variables, like groundwater availability, river discharge should also be included if data availability permits. At the same time, the feedback of socio-economic capacities to irrigation techniques share changes, should also be considered in future work.

Reviewer 2 Comment 4

In this study, it is assumed that the evolution of drip and sprinkler irrigation to replace flood irrigation is at a constant rate of 1%, and as you mentioned, paddy irrigation is included in flood irrigation, and planting rice with drip and sprinkler irrigation techniques seems technically challenging currently. Especially in East Asia, where rice cultivation is dominant, will this underestimate the irrigation impacts on extreme dry and humid heat in future scenarios?

Response We are sorry for the confusion. In this study, paddy irrigation is applied by default when it is rice, but when presenting the irrigation techniques share, it is counted as flood irrigation. We added some texts in the discussion.

L334-339

Moreover, in this baseline dataset, flood irrigation and paddy irrigation are not distinguished as separate categories. Instead, paddy irrigation is included under the broader classification of flood irrigation, despite the practical differences between the two methods in actual implementation. In the future, more information should be collected through the national agricultural census, and the classification methods to distinguish different irrigation techniques based on remote sensing data need to be developed.

Reviewer 2 Comment 5

CESM2 has no limits on irrigation withdrawals for the irrigation module, and in fact irrigation withdrawals are limited by the regional river flows, and for the future new hotspots mentioned in the manuscript, whether the unconstrained application of irrigation withdrawals would overestimate the irrigation impacts on the extreme heat in the region represented by the Middle East and Africa.

Response We thank Reviewer 2 for pointing this limitation out and we have added some text in the Discussion section.

L285-287

In this study, water availability is not activated, so the simulated IWW mostly indicates irrigation demand, and the requirement for water may therefore not be fulfilled, leaving crop yields in danger.

L349-351

Furthermore, precipitation is the only variable used to indicate hydro-climatic complexity, but some other variables, like groundwater availability, river discharge should also be included if data availability permits.

Reviewer 2 Comment 6

Lines 219-222 illustrate that irrigation practices in traditionally highly irrigated areas have minor impacts on extreme dry heat under the future scenarios, this point deserves more in-depth comparative analysis and discussion. The comparisons in Figures 4c and f seem to compare extreme dry heat differences between different SSP-RCP pathways rather than the impact of irrigation.

Response Thanks to Reviewer 2 for this suggestion. We added some text to highlight this point in the Discussion section.

L302-312

Under both SSP1-2.6 and SSP3-7.0, increased greenhouse gas emissions increase the frequency of extreme dry-heat events (Figure 6a,c). Irrigation has been proposed as a climate-effective land management strategy, and has been found to slow or even reverse local warming trends in recent decades. However, although the cooling impacts of irrigation on extreme dry-heat persist in projections (Figure 5e,g), we find that irrigation will no longer be able to create cooling islands in the future as effectively as it did in the past. This could be attributed to the pronounced impacts of climate change, as well as the upgrade in irrigation techniques over some traditional irrigation regions, as less water will be applied to fields. Notably, the effects of irrigation on dry-heat extreme frequency will be much smaller compared to the impact of following different SSP-RCPs (Figure 5e,g and 6e), in terms of both magnitude and spatial coverage.

We also restructured the Results section and removed the comparison between scenarios without irrigation (SSP1_NOI and SSP3_NOI).

Reviewer 2 Comment 7

As you mentioned in line 315, some areas that are currently considered unsustainable for irrigation practices continue to experience an increase in irrigation water withdrawal due to irrigation expansion under future scenarios, and their viability deserves further validation and discussion.

Response We admit this is a significant limitation in this study, and we therefore add some text in the Discussion section (see also Reviewer 2 Comment 5).

L362-365

To further improve irrigation representation in Earth system models, new research could aim at collecting and calibrating crop- and irrigation-related parameters regionally, implementing crop rotations and multiple cropping seasons, and incorporating groundwater availability and evolution.

Reviewer 2 Comment 8

This study is a continuation of the authors' recent achievement (DOI:10.1038/s41467-025-56356-1), which has in-depth insights in this field, and I believe that if they could consider how to guide the actual irrigation patterns towards the most beneficial direction for human based on the previous quantitative analysis and mechanism exploration studies of irrigation impacts on heat stress in the discussion section, it will provide a solid theoretical basis for the climate change adaptation strategies in agriculture.

Response Thanks to Reviewer 2 for this suggestion. We added some text in the Discussion section.

L251-258

In this study, we, for the first time, design ITS scenarios aligned with the SSP-RCPs, improve an Earth system model to incorporate this information, and perform fully-coupled climate simulations under varying climate and irrigation transition pathways. We then analyse the simulation outputs for irrigation water demand and irrigation-induced impacts on dry and moist-heat stress. The results exhibit a strong sensitivity to the choice of SSP-RCP scenarios, highlighting their important implications for future irrigation planning and development. While these scenarios rely on several idealised assumptions, the underlying narratives still offer valuable insights for guiding future irrigation strategies.

L291-295

Limiting unsustainable irrigation expansion and enhancing irrigation efficiency are, therefore, crucial for mitigating water scarcity. Achieving this requires a thorough assessment of current and projected hydrological conditions prior to selecting areas for cropland expansion, as well as the development and deployment of high-efficiency irrigation technologies, particularly in regions already facing water stress.

2.2 Minor comments

Reviewer 2 Comment 9

The abbreviation of Figure 4 on comparing the differences between future and historical scenarios fails to convey a clear enough meaning

Response Thanks for pointing it out. We have restructured the Results section and changed the captions. Please check the captions of Figure 5 and 6 in the main text.

Reviewer 2 Comment 10

Provide formulas for key evaluation indicators, such as wet bulb temperature, in the methodology section for better independent understanding.

Response Thanks for this suggestion. We have added it to Methods.

Reviewer 2 Comment 11

The figures and tables in the Appendix would be better as Support Information for the manuscript.

Response Thanks for the advice. We have moved them to the supplementary material.

Reviewer 2 Comment 12

Driving CESM2 simulations using SSP-RCP scenarios and innovative ITS datasets is the core of this study, and the necessary descriptions of the CESM2 modeling related setup in the Methods section (or support information) will make the paper more perfect.

Response Thank you for this advice. We have added some text in the Methods section.

L703-708

To allow three different irrigation techniques for each CFT per grid cell, we expand the cropland unit to 128 columns, one rainfed, one drip irrigated, one sprinkler irrigated, and one flood irrigated for each CFT. The calculations of the land fluxes are conducted individually over each column before being averaged to the grid cell level. Transient ITS is implemented through the preparation of input land use and management time series, in which we provide the annual information of ITS from the year 2015 to 2100.

L737-739

In CESM2, when transient land-use is activated, the surface map is updated yearly, and the ITS is incorporated in the surface map, which enables the model to use transient ITS.

3 Reviewer 3

3.1 Major comments

Reviewer 3 Comment 1

The study “Compounding future escalation of emissions- and irrigation-induced increases in humid-heat stress” discusses different irrigation scenarios and their impact on various heat events. It is a valuable contribution to understanding the feedback between irrigation methods and local heat stress. I thoroughly enjoyed reading the paper. I have a few comments that may help expand the discussion, which are listed below.

Response We thank Reviewer 3 for these supportive words and comments, which help us to improve the manuscript. Below, we address all comments one by one.

Reviewer 3 Comment 2

What if the assumed ITS transition rate of 1% per year is not accurate? Can you briefly describe what would happen if the ITS rate were different, and what the plausible scenarios might be?

Response Thanks for pointing this out. Indeed, this 1% per year rate is set based on the available irrigation techniques share time series from two countries, the USA and Iran, so it may be unrealistic to assign such a value for all countries. The default transition rate directly determines the time it takes for every country to achieve the maximum drip and sprinkler irrigation fraction, which is based on the distribution of local crop types, as shown in TAB 3.

We also added some discussion in the main text.

L341-353

Regarding the assumptions applied for transitions in ITS, only socioeconomic and hydro-climatic conditions are considered as drivers. In reality, farmers’ individual decision-making and government policies are not fully determined by socio-economic capacities. For example, farmers’ decisions to improve irrigation efficiency are motivated by multiple factors, especially the cost and benefits. Another issue is that the socio-economic capacity variables are assumed to be the same within each country, ignoring the internal variability within bigger countries. Moreover, a 1% default rate of flood irrigation reduction is assumed for all countries based on historical time series of two countries (see Method), which may not be realistic for other countries, and changing this default rate may lead to different results.

TAB 3: Suitability of irrigation techniques of each crop type

Crop type	drip	sprinkler	flood
Spring wheat	No	Yes	Yes
Winter wheat	No	Yes	Yes
Barley	No	Yes	Yes
Winter barley	No	Yes	Yes
Rye	No	Yes	Yes
Winter rye	No	Yes	Yes
Rice	No	No	Yes
Temperate maize	No	Yes	Yes
Tropical maize	No	Yes	Yes
Millet	No	Yes	Yes
Sorghum	No	Yes	Yes
Pulses	Yes	Yes	Yes
Cassava	No	No	No
Sunflower	Yes	Yes	Yes
Temperate soybean	Yes	Yes	Yes
Tropical soybean	Yes	Yes	Yes
Groundnuts	No	Yes	Yes
Rapeseed	No	Yes	Yes
Sugarcane	No	Yes	Yes
Citrus	Yes	Yes	Yes
Cocoa	Yes	Yes	Yes
Coffee	Yes	Yes	Yes
Cotton	Yes	Yes	Yes
Date palm	Yes	Yes	Yes
Grapes	Yes	Yes	Yes
Oil palm	Yes	Yes	Yes
Potatoes	Yes	Yes	Yes
Fodder grass	No	Yes	Yes
Miscanthus	No	Yes	Yes

Reviewer 3 Comment 3

Doubling the number of cropland columns from 64 to 128 increases model complexity and computational cost. If the data used to allocate irrigation methods across columns is uncertain or based on assumptions, how do you ensure that this added complexity results in more reliable outcomes? Can you also discuss the limitations of the model used?

Response Yes this is a good point. Although the land model consumes fewer computational resources than the atmosphere model, increasing the number of basic calculation units still increases its complexity. In this study, we expanded the model to represent time-varying irrigation techniques share, but whether it is still worth it requires evaluation and validation. Thus, we add some texts to the discussion.

L365-369

At the same time, the increased complexity of the model in this study (cropland columns expanded from 64 to 128) requires more computational resources, underscoring the importance of evaluating the new module on reproducing surface fluxes. Previous evaluations were mainly focused on a single-point scale, and in the future, the validation at regional and local scales should be conducted.

Reviewer 3 Comment 4

Could you elaborate on the assumptions underlying the hydro-economic framework? For instance: i) Classifying countries by national-level variables misses within-country variability in both irrigation practices and driving factors ii) Since the driver selection is based on Spearman's correlation, how do you address the fact that correlation does not imply causation? iii) Why is precipitation used as the sole indicator of hydro-climatic conditions, and how do you account for other key factors like groundwater availability, river discharge, infrastructure, or water rights? iv) The framework is static and scenario-based: it projects future ITS based on fixed drivers but does not simulate feedbacks (e.g., how changes in ITS might influence GDP, water stress, or governance over time).

Response Sorry for not clearly explaining the whole process of irrigation techniques share projection and the lack of a comprehensive discussion regarding this.

(1) Indeed, the internal variabilities of socio-economic variables are ignored, so we added some texts in the Discussion section.

L345-346

Another issue is that the socio-economic capacity variables are assumed to be the same within each country, ignoring the internal variability within bigger countries.

ii) Indeed, correlation does not imply causation, but we do believe that higher socio-economic capacities are beneficial for irrigation techniques upgrade. Of course, the dataset generated

here is highly idealised, but the plan is to generate a dataset aligning with the narratives under the SSP framework.

We added a paragraph in the Discussion section.

L260-273

The assumptions regarding land use and agriculture under SSP1 and SSP3 are provided as "strong regulations to avoid environmental tradeoffs; improvements in agriculture productivity; rapid diffusion of best practices", and "Hardly any regulation; continued deforestation due to competition over land and rapid expansion of agriculture; Low technology development, restricted trade". The LUH2 dataset agrees with those narratives, with the key feature that AEI keeps almost constant under SSP1-2.6 and expands rapidly under SSP3-7.0 (Figure 1, 2). Some assumptions regarding the irrigation water use efficiency have also been proposed, with "large improvements in irrigation water use efficiency where possible" for SSP1 and "only modest improvements in irrigation water use efficiency" for SSP3. The generated ITS data align well with those assumptions, with a rapid decrease in traditional inefficient flood irrigation under SSP1-2.6 and a slower decrease under SSP3-7.0 (Figure 1, 3). Although we apply a simplified adjustment factor to account for annual ITS changes based on socio-economic and hydro-climatic variables (see Methods), the resulting estimates are consistent with prevailing assumptions, thereby offering a valuable enhancement to existing datasets.

iii) Because precipitation is the only variable which shows significant correlation with irrigation techniques in this study (Table S2). Other factors are not quantified in this study, at least not available from ISIMIP simulations, but we believe that infrastructure and water rights can be included in the socio-economic capacities in future.

We therefore added a sentence in the Discussion section.

L349-351

Furthermore, precipitation is the only variable used to indicate hydro-climatic complexity, but some other variables, like groundwater availability, river discharge should also be included if data availability permits.

(iv) Yes, there is no feedback included, as socioeconomic scenario is typically treated as an input to the ESM and thus not explicitly modelled. However, these feedbacks should be considered in future work where a closer interaction of ESMs and IAMs could be explored. We therefore added a sentence in the Discussion section.

L351-353

At the same time, the feedback of socio-economic capacities to irrigation techniques share changes, should also be considered in future work.

References

- [1] O'Neill, B., Kriegler, E., Ebi, K., Kemp-Benedict, E., Riahi, K., Rothman, D., Van Ruijven, B., Van Vuuren, D., Birkmann, J., Kok, K. & Others The roads ahead: Narratives for shared socioeconomic pathways describing world futures in the 21st century. *Global Environmental Change*. **42** pp. 169-180 (2017)
- [2] Wada, Y., Flörke, M., Hanasaki, N., Eisner, S., Fischer, G., Tramberend, S., Satoh, Y., Van Vliet, M., Yillia, P., Ringler, C. & Others Modeling global water use for the 21st century: The Water Futures and Solutions (WFaS) initiative and its approaches. *Geoscientific Model Development*. **9**, 175-222 (2016)
- [3] Jägermeyr, J., Gerten, D., Heinke, J., Schaphoff, S., Kummu, M. & Lucht, W. Water savings potentials of irrigation systems: global simulation of processes and linkages. *Hydrology And Earth System Sciences*. **19**, 3073-3091 (2015)

Compounding future escalation of emissions- and irrigation-induced increases in humid-heat stress

August 12, 2025

Yi Yao^{*1,2}, Yusuke Satoh³, Nicole van Maanen⁴, Sabin Taranu¹, Jessica Keune⁵, Steven J. De Hertog^{6,1}, Seppe Lampe¹, David M. Lawrence⁷, William J. Sacks⁷, Yoshihide Wada⁸, Agnès Ducharme⁹, Benjamin I. Cook^{10,11}, Sonia I. Seneviratne², Laibao Liu^{12,13}, Jonathan R. Buzan¹⁴, Jonas Jägermeyr^{15,10,17}, Wim Thiery¹
yi.yao@vub.be

¹ Department of Water and Climate, Vrije Universiteit Brussel, Brussels, Belgium

² Institute for Atmospheric and Climate Science, Eidgenössische Technische Hochschule Zürich, Zurich, Switzerland

³ Japan Agency for Marine-Earth Science and Technology, Yokohama, Japan

⁴ Department of Water and Climate Risk, Institute for Environmental Studies (IVM), Vrije Universiteit Amsterdam

⁵ Forecast and Services Department, European Center for Medium-Range Weather Forecasts, Bonn, Germany

⁶ Q-ForestLab, Department of Environment, Universiteit Gent, Ghent, Belgium

⁷ NSF National Center for Atmospheric Research, Boulder, Colorado, USA

⁸ Biological and Environmental Science and Engineering Division, King Abdullah University of Science and Technology (KAUST), Saudi Arabia

⁹ Laboratory 7619 METIS, Sorbonne Université, CNRS, EPHE, IPSL, Paris, France

¹⁰ NASA Goddard Institute for Space Studies, New York, New York

¹¹ Lamont-Doherty Earth Observatory, Palisades, New York

¹² Department of Geography, The University of Hong Kong (HKU), Hong Kong

¹³ Institute for Climate and Carbon Neutrality, The University of Hong Kong (HKU), Hong Kong

¹⁴ Department of Chemistry and Bioscience, Aalborg University, Aalborg, Denmark

¹⁵ Columbia University, Climate School, New York, USA

¹⁶ Potsdam Institute for Climate Impact Research (PIK), Member of the Leibniz Association, Potsdam, Germany

Contents

1	Reviewer 1	4
1.1	Major comments	4
	Reviewer 1 Comment 1	4
1.2	Minor comments	4
	Reviewer 1 Comment 2	4

Reviewer 1 Comment 3	4
Reviewer 1 Comment 4	4
Reviewer 1 Comment 5	4
Reviewer 1 Comment 6	5
Reviewer 1 Comment 7	5
Reviewer 1 Comment 8	5
Reviewer 1 Comment 9	6
Reviewer 1 Comment 10	7

Abstract

This response letter contains numbered figures and tables and references to them. To prevent confusion, the figures and tables embedded within this response letter are called FIGs and TABs, while the figures and tables embedded within the manuscript are still called Figures and Tables. The following convention is applied to denote modification in the original manuscript: new text.

1 Reviewer 1

1.1 Major comments

Reviewer 1 Comment 1

The manuscript has been revised and many improvements have been made. This includes a restructuring and better description of the results. Moreover, the discussion section has been revised and restructured. These changes have increased the readability of the manuscript. Apart from the minor comments below, I think the manuscript is acceptable for publication.

Below I have provided specific comments to the text, figures and tables.

Response We thank Reviewer 1 for these supportive words and the constructive suggestions, which has helped us to improve our manuscript. Below, we address every comment carefully and explain the corresponding changes in the manuscript.

1.2 Minor comments

Reviewer 1 Comment 2

Lines 41-45: Please revise this sentence. I think the previous version said correctly “increases”, which now was changed to “increase” (line 42). Moreover, it is unclear to what “them” refers to in line 43.

Response We thank Reviewer 1 for finding this mistake. We have revised this sentence.

Abstract, L40-43

Moreover, moist-heat extreme event frequency **increases** more substantially (by ≥ 1600 hours yr^{-1} under SSP3-7.0 in tropical regions), and irrigation further **amplifies the hours of exposure** (for example by ≥ 100 hours yr^{-1} in South Asia).

Reviewer 1 Comment 3

Lines 93-96: I still don’t understand why the authors consider the SSP5 scenario here. The main document does not include any reference to this scenario, so why mention it here? Again, I highly suggest to remove the SSP5 scenario altogether from the manuscript (also from the Methods section).

Response We thank Reviewer 1 for this suggestion. We have now completely removed the contents regarding SSP5-8.5.

Reviewer 1 Comment 4

Line 180: Please replace “is increased” by “has increased”.

Response We thank Reviewer 1 for finding this mistake. We have corrected it.

Reviewer 1 Comment 5

Line 182: I'm not sure if "intensify" is correctly used here. Maybe better say "increase" or "amplify".

Response We thank Reviewer 1 for this suggestion. We have replaced 'intensify' by 'amplify' in this sentence, and checked throughout the entire manuscript to make sure that this word is not misused.

Reviewer 1 Comment 6

Lines 241-244: These results consider the traditional hotspot areas? Please indicate in the text.

Response We thank Reviewer 1 for this suggestion. We add some text to this sentence.

Results, L238-240

Over traditional hotspot areas, Irrigation-induced cooling effects on dry-heat and concurrent warming effects on moist-heat contribute to an amplified divergence in exposure hours between these two types of extreme events.

Reviewer 1 Comment 7

Lines 249-250: I'm not so convinced about the difference between the scenarios with and without irrigation. In Figure 7 these differences are not that big for most of the data shown. Only for dry-heat for the traditional irrigation hotspots this difference seems rather obvious, but for the other scenarios not that much. Please change this sentence accordingly

Response We thank Reviewer 1 for this suggestion. We have revised this sentence.

Results, L246-247

Furthermore, over intensely irrigated areas, irrigation plays an important role as an anthropogenic climate forcing, which should not be neglected in future studies.

Reviewer 1 Comment 8

Lines 260-264: Please explicitly indicate which assumption belongs to which SSP scenario.

Response We thank Reviewer 1 for this suggestion. We have revised this sentence.

Discussion, L248-252

The assumptions regarding land use and agriculture under SSP1 and SSP3 are provided as "SSP1, a green road; strong regulations to avoid environmental tradeoffs; improvements in agriculture productivity; rapid diffusion of best practices", and "SSP3, a rocky road; hardly any regulation; continued deforestation

due to competition over land and rapid expansion of agriculture; Low technology development, restricted trade" (1).

Reviewer 1 Comment 9

Lines 327-369: These three paragraphs include many limitations to the presented methods. The authors propose some changes to the existing method. However, it would be interesting to read how these proposed methods would affect the results. So, instead of including a longer list of improvements, it would be interesting to select the most relevant and describe how these improvements might affect the results.

Response We thank Reviewer 1 for this suggestion and have added one sentence for each paragraph to indicate the effects of these improvements.

Discussion, L316-314

In this study, we estimate the changes in irrigation techniques under socioeconomic and greenhouse gas concentration scenarios, drawing on the differences in socioeconomic development stage and baseline aridity level of every country (see Methods). However, uncertainties remain in this dataset because of both the lack of observational data and the assumptions applied. The baseline global gridded dataset for present-day irrigation techniques distribution (2) is not a directly observed dataset. Instead, it was generated using a decision tree algorithm based on country-level data from multiple sources (3; 4; 5), further informed by global cropland maps. Moreover, in this baseline dataset, flood irrigation and paddy irrigation are not distinguished as separate categories. Instead, paddy irrigation is included under the broader classification of flood irrigation, despite the practical differences between the two methods in actual implementation. In future, more information should be collected through the national agricultural census, and the classification methods to distinguish different irrigation techniques based on remote sensing data need to be developed. This will provide a more realistic basis for projecting the distribution of irrigation techniques and will facilitate the understanding of key factors for improving irrigation efficiency.

Regarding the assumptions applied for transitions in ITS, only socioeconomic and hydro-climatic conditions are considered as drivers. In reality, farmers' individual decision-making and government policies are not fully determined by socio-economic capacities. For example, farmers' decisions to improve irrigation efficiency are motivated by multiple factors (6), especially the cost and benefits (7). Another issue is that the socio-economic capacity variables are assumed to be the same within each country, ignoring the internal variability within bigger countries. Moreover, a 1% default rate of flood irrigation reduction is assumed for all countries based on historical time series of two countries (see Method), which may not be realistic for other countries, and changing this default rate may lead to different results. Furthermore, precipitation is the only variable used to indicate hydro-climatic complexity, but some other variables, like groundwater availability, river discharge, should also be included if data availability permits. At the same time, the feedback of socio-economic capacities to irrigation techniques share changes, should also be considered in future work. This will sup-

port the development of more realistic datasets on the share of irrigation techniques consistent with future pathways, enabling more accurate predictions of the changes in irrigation practice.

The CESM2 (8) was recently expanded to represent various irrigation techniques (9), and we here extend this functionality to capture transient changes in these techniques. Although the implementation of multiple irrigation techniques has led to improved performance and utility (9), limitations remain in the model's crop and irrigation representation. For example, the CESM2 has a globally identical crop and irrigation technique parameterisation for each crop type, and only one single cropping season is allowed per year. Furthermore, in this study, the limitation of water withdrawal for irrigation is not activated, as there is currently no implementation of groundwater availability. To further improve irrigation representation in Earth system models, new research could aim at collecting and calibrating crop- and irrigation-related parameters regionally, implementing crop rotations and multiple cropping seasons, and incorporating groundwater availability and evolution. At the same time, the increased complexity of the model in this study (cropland columns expanded from 64 to 128) requires more computational resources, underscoring the importance of evaluating the new module on reproducing surface fluxes. Previous evaluations were mainly focused on a single-point scale (9), and in the future, the validation at regional and local scales should be conducted. These efforts will enable accurate simulation of irrigation-induced impacts on the Earth system while minimizing unnecessary consumption of computational resources.

Reviewer 1 Comment 10

Lines 371-376: I suggest to end the Discussion section with a paragraph with some concluding remarks, instead of the limitations of the study.

Response We thank Reviewer 1 for this suggestion. We have moved the first paragraph of the Discussion section to the end and added some text.

Discussion, L373-388

In this study, we, for the first time, design ITS scenarios aligned with the SSP-RCPs, improve an Earth system model to incorporate this information, and perform fully-coupled climate simulations under varying climate and irrigation transition pathways. We then analyse the simulation outputs for irrigation water demand and irrigation-induced impacts on dry and moist-heat stress. The results exhibit a strong sensitivity to the choice of SSP-RCP scenarios, highlighting their important implications for future irrigation planning and development. While these scenarios rely on several idealised assumptions, the underlying narratives still offer valuable insights for guiding future irrigation strategies. More specifically, under SSP1-2.6, conservative irrigation expansion combined with rapid adoption of improved techniques can reduce irrigation water withdrawals, whereas under SSP3-7.0, massive expansion coupled with slow technological upgrades substantially increases withdrawals, generating new irrigation hotspot regions, particularly in Africa. As a scenario with higher greenhouse gas con-

centrations, SSP3–7.0 results in more frequent dry- and moist-heat extremes than SSP1–2.6, with the increase in moist-heat exposure being more pronounced. Irrigation—particularly in intensely irrigated areas—mitigates dry-heat extremes but slightly amplifies moist-heat, emerging as an important climate forcing that should not be overlooked in future studies.

References

- [1] O'Neill, B., Kriegler, E., Ebi, K., Kemp-Benedict, E., Riahi, K., Rothman, D., Van Ruijven, B., Van Vuuren, D., Birkmann, J., Kok, K. & Others The roads ahead: Narratives for shared socioeconomic pathways describing world futures in the 21st century. *Global Environmental Change*. **42** pp. 169-180 (2017)
- [2] Jägermeyr, J., Gerten, D., Heinke, J., Schaphoff, S., Kummu, M. & Lucht, W. Water savings potentials of irrigation systems: global simulation of processes and linkages. *Hydrology And Earth System Sciences*. **19**, 3073-3091 (2015)
- [3] FAO AQUASTAT - FAO's Global Information System on Water and Agriculture. <https://www.fao.org/aquastat/en/overview/>. (2023)
- [4] ICID International Commission on Irrigation and Drainage: Sprinkler and Micro Irrigated area. (2012)
- [5] Rohwer, J., Gerten, D. & Lucht Development of functional irrigation types for improved global crop modelling. (Potsdam Institute for Climate Impact Research, 2007)
- [6] Skaggs, R. & Samani, Z. Farm size, irrigation practices, and on-farm irrigation efficiency. *Irrigation And Drainage: The Journal Of The International Commission On Irrigation And Drainage*. **54**, 43-57 (2005)
- [7] Tai, X., Feng, F. & Sun, F. Farmers' Willingness and Adoption of Water-Saving Agriculture in Arid Areas: Evidence from China. *Sustainability*. **16**, 8112 (2024)
- [8] Danabasoglu, G., Lamarque, J., Bacmeister, J., Bailey, D., DuVivier, A., Edwards, J., Emmons, L., Fasullo, J., Garcia, R., Gettelman, A. & Others The community earth system model version 2 (CESM2). *Journal Of Advances In Modeling Earth Systems*. **12**, e2019MS001916 (2020)
- [9] Yao, Y., Vanderkelen, I., Lombardozzi, D., Swenson, S., Lawrence, D., Jägermeyr, J., Grant, L. & Thiery, W. Implementation and evaluation of irrigation techniques in the community land model. (Fort Collins, Colo.: [Verlag nicht ermittelbar], 2022)